# DiffuMamba: High-Throughput Diffusion LMs with Mamba Backbone

Vaibhav Singh [1 2 3 *]  Oleksiy Ostapenko [1]  Pierre-André Noël [1]  Eugene Belilovsky [2 3]  Torsten Scholak [1]

## Abstract

Diffusion language models (DLMs) have emerged as a promising alternative to autoregressive (AR) generation, yet their reliance on Transformer backbones limits inference efficiency due to quadratic attention or KV-cache overhead. We introduce *DiffuMamba*, a masked diffusion language model built on a bidirectional Mamba backbone that combines the diffusion objective with linear-time sequence modeling, and *DiffuMamba-H*, a hybrid variant with interleaved attention. Across scales up to 1.3B parameters, our models match Transformer-based diffusion in downstream performance while achieving up to **8.2×** and **4.3×** higher inference throughput, respectively, on long sequences. We further present a systematic analysis of inference efficiency across modern DLM variants combining asymptotic complexity with empirical measurements. Notably, cache-efficient block diffusion with Mamba mixers emerges as the only strategy that scales linearly with sequence length and achieves the strongest performance across all baselines, suggesting a promising direction for future diffusion-based generation systems.

## 1. Introduction

The majority of the state-of-the-art Large Language Models (LLMs) (Brown et al., 2020; Achiam et al., 2023; Chowdhery et al., 2023; Touvron et al., 2023a) rely on full multi-head attention mechanism (MHA) (Bahdanau et al., 2015; Vaswani et al., 2017) and are trained with the *autoregressive* (AR) objective: a Transformer predicts the next token given all previous tokens. While simple and powerful, this paradigm faces several persistent limitations. First, AR

---

*Work done during a research internship at ServiceNow Research. [1]ServiceNow Research, Montreal, Canada [2]Mila [3]Concordia University. Correspondence to: Vaibhav Singh <vaibhav.singh@mila.quebec>, Oleksiy Ostapenko <oleksiy.ostapenko@servicenow.com>.

*Proceedings of the $43^{rd}$ International Conference on Machine Learning*, Seoul, South Korea. PMLR 306, 2026. Copyright 2026 by the author(s).

decoding is inherently sequential: every generated token depends on the full prefix, and thus inference latency grows linearly with output length. The AR training paradigm also limits data efficiency: each training example provides supervision for only a single next-token prediction, constraining the amount of learning signal that can be extracted from limited data (Ni et al., 2025a). This limitation becomes increasingly relevant as high-quality data, rather than compute, emerges as a key bottleneck in scaling language models. Furthermore, AR Transformers are constrained by both memory and compute: the KV cache grows linearly with context length, inducing increasing memory pressure, while the attention mechanism incurs quadratic compute cost in the sequence length, limiting long-context inference and training (Zhou et al., 2024; Tay et al., 2021). These constraints limit throughput and test-time scaling efficiency, motivating research into alternative architectures.

Diffusion Language Models (DLMs) (Li et al., 2025) provide a flexible alternative to AR generation. Unlike AR decoders, DLMs iteratively denoise entire corrupted sequences in parallel, enabling non-sequential multi-token generation, partial infilling, and self-correction. In principle, this opens the door to faster generation and self-refined reasoning, especially when fewer denoising steps suffice for acceptable quality. However, a key bottleneck persists: *all current DLMs rely on Transformer backbones*, making their iterative denoising compute or/and memory intensive pushing their throughput well below the AR competitors especially at long sequences. In standard DLMs, denoising proceeds via repeated full sequence re-encoding using bidirectional attention, as each step conditions on both past and future tokens. Because token states evolve across denoising steps, representations cannot be incrementally reused, leading to per-step costs that scale quadratically with sequence length. While several DLM variants enable KV caching (Wu et al., 2026; Ma et al., 2026; Nguyen-Tri et al., 2025; Fathi et al., 2025), the core architectural limitation remains: *as the cache grows with sequence length, it induces increasing memory traffic becoming the key bottleneck at inference.* Consequently, while inference cost necessarily scales with the number of denoising steps and the sequence length, Transformer-based DLMs incur additional attention and cache related overheads making per token latency grow with sequence length. The result is a paradox: *DLMs*

*promise flexible text generation, yet their efficiency is constrained by increasing memory overhead and periodic cache recomputation induced by the Transformer backbone (Wu et al., 2026).*

In parallel with advances in diffusion language modeling, state-space models (SSMs) have emerged as a powerful class of sequence mixers with linear-time complexity (Gu et al., 2022; Poli et al., 2023; Fu et al., 2023; Gu & Dao, 2024; Dao & Gu, 2024). Recent studies demonstrate that SSMs and SSM–Transformer hybrids can match or even outperform Transformers while achieving substantially higher inference throughput (Gu & Dao, 2024; Somvanshi et al., 2025; Wang et al., 2025a). Despite their potential, SSMs remain unexplored in DLMs. This raises a key question: *can structured recurrence serve as an effective language denoiser while enabling faster inference?* Our central motivation is to explore this intersection of efficient SSM backbones and masked discrete diffusion for language modeling.

To this end, we pre-train standard full-sequence DLMs (Sahoo et al., 2024) whose Transformer backbone is modified in two ways: (i) by replacing all Multi-Head Attention (MHA) mixers with Mamba-2 (Dao & Gu, 2024), and (ii) by replacing only a subset of MHA mixers with Mamba-2 to form a hybrid architecture. Further, we employ bidirectional Mamba-2 mixers, as masked diffusion requires conditioning on both past and future context at each denoising step. By removing quadratic attention, we reduce per-step latency and memory pressure without altering the probabilistic semantics of masked diffusion. As a result, Mamba-based DLMs achieve higher throughput than both Transformer-based DLMs and autoregressive baselines across a range of decoding algorithms, including block diffusion (Arriola et al., 2025; Wu et al., 2026).

The main contributions of this work are:

- **New architectural direction.** We propose *Diffu-Mamba*, which replaces Transformer denoisers with bidirectional Mamba-2 mixers for discrete masked diffusion language modeling, and *DiffuMamba-H*, a sparse hybrid that interleaves one attention block every four Mamba blocks ($\approx 20\%$ attention). Together they demonstrate that iterative denoising does not require dense attention, positioning linear-time backbones as a scalable alternative for DLMs.

- **Controlled evaluation across scales, contexts, and benchmarks.** We compare `DiffuMamba` and `DiffuMamba-H` against `DiffuTran` under identical training data, tokenization, noise schedules, and decoding steps at three parameter budgets (240M, 0.5B, 1.3B). `DiffuMamba-H` matches or surpasses `DiffuTran` on Val. PPL, zero-shot PPL, downstream accuracy, Gen PPL, and MAUVE, and ex-

trapolates cleanly to $2\times$ the training context where `DiffuTran`'s Avg. PPL collapses by 76%.

- **Comprehensive throughput and compute analysis.** We present an asymptotic and empirical analysis of DLM inference strategies, scaling generation length beyond 100k tokens. Mamba-backed DLMs achieve up to **8.2×** throughput in full-sequence denoising (Figures 2a and 2b) and **2.3×** in block-wise autoregressive denoising (Figures 2c and 2d) over `DiffuTran`, with the advantage holding at $B{=}8$ and under block-cache decoding (best Gen PPL coincides with the best-throughput block size). A per-layer FLOPs analysis confirms these gains come from architecture, not parameter count: at $L{=}8$K, `DiffuMamba` uses only $0.82\times$ `DiffuTran`'s FLOPs despite carrying $1.30\times$ the parameters.

## 2. Related Work

**Diffusion Language Models (DLMs)** Diffusion modeling replace left-to-right decoding with iterative denoising. While early work focused on continuous domains (Ho et al., 2020; Song et al., 2021; Rombach et al., 2022; Peebles & Xie, 2023), recent research has adapted diffusion to *discrete* text. Austin et al. (2021) introduced structured transition matrices with absorbing states, and DiffusionBERT (He et al., 2023) employed a masked corruption process paired with a BERT-style denoiser. Subsequent studies have further refined masking strategies, noise schedules, and sampling procedures (Chen et al., 2023; Lou et al., 2024; Varma et al., 2025; Fathi et al., 2025). Masked diffusion approaching autoregressive quality (Sahoo et al., 2024; 2025) enabled scalable models (Ye et al., 2025; Nie et al., 2026) competitive with LLaMA (Touvron et al., 2023b), with further gains from AR adaptation and instruction tuning (Gong et al., 2025; Zhu et al., 2025). To enhance inference efficiency, recent acceleration techniques (Ma et al., 2026; Wu et al., 2026; Liu et al., 2025b; Israel et al., 2026; Wang et al., 2025b) use approximate KV-caching mechanisms to speed up DLM inference. In contrast, `DiffuMamba-H` achieves its speedups without relying on KV caching, though its attention layers remain compatible with these optimizations.

**State-Space Models (SSMs) for Sequence Modeling** SSMs introduce structured recurrences computable via fast convolutions or scans, achieving linear-time scaling with sequence length. Early models like S4 (Gu et al., 2022) demonstrated strong long-context capacity, while Hyena (Poli et al., 2023) and H3 (Fu et al., 2023) explored alternative structured operators for language modeling. Mamba (Gu & Dao, 2024) extended this line by introducing input-conditioned selective state spaces, matching or surpassing Transformers in accuracy with lower latency and memory

use. Somvanshi et al. (2025) further survey the breadth of SSM architectures and highlight their efficiency advantages. In autoregressive language modeling, SSMs rival Transformers at comparable scales while offering higher tokens-per-second throughput. More recent works have evolved various linear-complexity LMs and hybrids (Wang et al., 2025a; Ostapenko et al., 2025), ranging from vector recurrences (De et al., 2024) to advanced gating mechanisms (Yang et al., 2025). Although SSM backbones have been used in diffusion models for non-textual domains such as generative modelling in biological sequences (Sahoo et al., 2024; Schiff et al., 2024), text diffusion LMs continue to rely on full-attention denoisers, leaving open whether SSM recurrences can replace attention for iterative denoising.

**Diffusion with Mamba/SSMs in Vision.** In vision, several studies have replaced Transformer-based DiT backbones with state-space models (SSMs), particularly Mamba variants (Gu & Dao, 2024; Ergasti et al., 2025; Dang et al., 2024), to improve diffusion efficiency. The Diffusion Mamba (DiM) family (Teng et al., 2024; Mo, 2025) achieves high-resolution image synthesis with multidirectional scanning and local feature enhancement, yielding notable throughput gains over attention-based models. VM-DDPM (Ju & Zhou, 2024) fuses convolutional locality with SSM-based global modeling to enhance structural fidelity in medical image generation. ZigMa (Hu et al., 2024) introduces zigzag scanning for faster, more memory-efficient diffusion while maintaining competitive quality, and Phung et al. (2024) incorporates wavelet transforms to strengthen local inductive biases, achieving faster convergence and favorable quality–efficiency trade-offs. Beyond diffusion, Zhu et al. (2024) generalizes Mamba as a vision backbone, and recent surveys (Xu et al., 2024; Liu et al., 2025a; Ergasti et al., 2025; Wang et al., 2024; Dang et al., 2024) map its rapid adoption across segmentation, restoration, and dense prediction. Collectively, these works show that replacing attention with Mamba in multi-step image diffusion preserves or improves quality while substantially lowering per-step compute.

## 3. Method

### 3.1. Preliminary

Masked Diffusion Models (MDMs) employ a forward noising process where tokens are progressively replaced by a special [MASK] token (Sahoo et al., 2024; Shi et al., 2024). This process is defined by the transition probability

$$q_{t|0}(\mathbf{x}_t \mid \mathbf{x}_0) = \prod_{i=1}^{L} q_{t|0}(x_t^i \mid x_0^i)$$
$$= \prod_{i=1}^{L} \text{Cat}\Big(x_t^i; \ (1-t)\delta_{x_0^i} + t\,\delta_{\text{MASK}}\Big), \quad (1)$$

where $t \in [0, 1]$ controls interpolation between the original data $\mathbf{x}_0$ (at $t = 0$) and a fully masked sequence (at $t = 1$). $\text{Cat}(\cdot)$ denotes the categorical distribution. A parametric model $p_\theta$ learns the reverse denoising process, and generation starts from all [MASK] and iteratively unmasks by sampling $p_\theta(x_0^i|\mathbf{x}_t)$.

Recent theory (MDM (Shi et al., 2024; Sahoo et al., 2024), RADD (Ou et al., 2025)) simplifies training from a variational bound to a reweighted cross-entropy over masked positions

$$\mathcal{L}_{\text{MDM}} = \int_0^1 \frac{1}{t} \mathbb{E}_{q_{t|0}(\mathbf{x}_t|\mathbf{x}_0)} \left[ \sum_{i:x_t^i=\text{MASK}} -\log p_\theta(x_0^i|\mathbf{x}_t) \right] dt. \quad (2)$$

This formulation scales to LLMs as DLMs, with LLaDA (Nie et al., 2026) and Dream-7B (Ye et al., 2025) matching autoregressive performance while enabling parallel decoding and flexible infilling.

The analytical reverse of the forward process defined in Equation 1 is computationally inefficient for generation, as it typically modifies only a single token at each step (Campbell et al., 2022; Lou et al., 2024). A common strategy to accelerate sampling is to employ an *MCMC-based approximation of the reverse process* (Nie et al., 2026), enabling the model to update multiple masked tokens in a single step when transitioning from a noise level $t$ to an earlier level $s < t$. This yields the following factorized form:

$$q_{s|t} = \prod_{i=1}^{L} q_{s|t}(\boldsymbol{x}_s^i \mid \boldsymbol{x}_t).$$

$$= \begin{cases} 1 & \text{if } \boldsymbol{x}_s^i = \boldsymbol{x}_t^i \neq [\text{MASK}], \\ \dfrac{s}{t} & \text{if } \boldsymbol{x}_s^i = \boldsymbol{x}_t^i = [\text{MASK}], \\ \dfrac{t-s}{t} q_{0|t}(\boldsymbol{x}_s^i \mid \boldsymbol{x}_t) & \text{if } \boldsymbol{x}_t^i = [\text{MASK}] \neq \boldsymbol{x}_s^i. \end{cases} \quad (3)$$

Here, $q_{0|t}(\boldsymbol{x}_s^i \mid \boldsymbol{x}_t)$ denotes the model-provided distribution over the vocabulary for predicting a non-[MASK] token when $\boldsymbol{x}_t^i$ is masked. In conditional generation settings e.g., generating a response $\boldsymbol{x}_0$ given a prompt $p$, the reverse diffusion process in Equation 3 must be adapted. In this case, the model's predictive distribution for unmasking a token becomes *prompt-conditioned*: $q_{0|t}(\boldsymbol{x}_s^i \mid \boldsymbol{x}_t, p)$, reflecting that token predictions now depend on both the intermediate noised sequence and the conditioning prompt.

### 3.2. Diffusion Mamba Language Models

We propose DiffuMamba, an MDM whose denoiser replaces the Transformer encoder with a *bidirectional state-space Mamba (BiMamba)* backbone (Zhu et al., 2024; Sahoo et al., 2024; Schiff et al., 2024), preserving the probabilistic structure of masked discrete diffusion while enabling

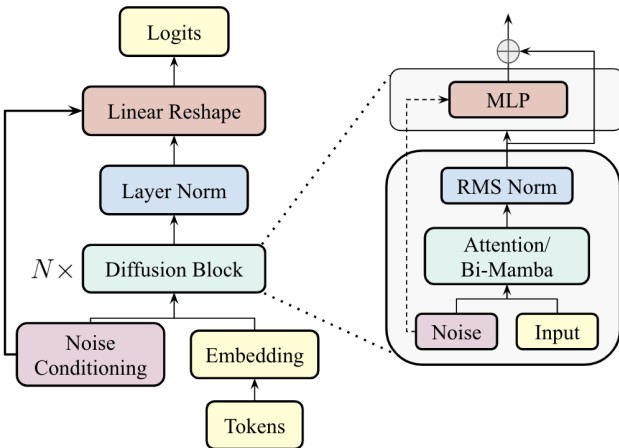

*Figure 1.* Schematic diagram of our proposed `DiffuMamba` architecture where mixer blocks replaces attention layers with bidirectional Mamba layers. For `DiffuMamba-H` we have interleaved attention layers after every $N-1$ Mamba layers. Attention provides global token interactions while Mamba enables efficient state space sequence modeling, allowing the hybrid denoiser to capture both long-range dependencies and local temporal dynamics with significantly improved efficiency. In our experiments, we fix $N=5$.

*linear-time* inference and substantially reduced memory overhead during multi-step denoising. We refer to transformer based DLM in our work as `DiffuTran` for simplicity.

**Architecture details.** We now describe the BiMamba denoiser architecture used in `DiffuMamba` also shown in Figure 1. Let $\mathbf{x} \in \mathbb{R}^{B \times L \times d}$ denote a batch of token embeddings, where $B$ is the batch size, $L$ the sequence length, and $d$ the hidden dimension. Each block is composed of two independent Mamba layers: one processes the sequence in the forward direction, and the other processes the sequence in the reverse direction

$$\mathbf{h}_i^{\rightarrow} = A_f * \mathbf{h}_{i-1}^{\rightarrow} + B_f * \mathbf{x}_i, \quad i = 1, \dots, L, \quad (4)$$
$$\mathbf{h}_i^{\leftarrow} = A_b * \mathbf{h}_{i+1}^{\leftarrow} + B_b * \mathbf{x}_i, \quad i = L, \dots, 1. \quad (5)$$

Here $A_f, B_f, A_b, B_b \in \mathbb{R}^{k \times d}$ are learnable state-transition kernels implemented as 1D causal and anti-causal convolutions or efficient scan operations (Gu & Dao, 2024). The two directional feature streams are fused through simple *additive integration*

$$\text{Mamba}(x_i) = \mathbf{h}_i = \mathbf{h}_i^{\rightarrow} + \mathbf{h}_i^{\leftarrow}, \quad (6)$$

providing a symmetric context representation while maintaining numerical stability. The resulting hidden sequence $\mathbf{H} = (\mathbf{h}_1, \dots, \mathbf{h}_L)$ is then normalized and passed through a lightweight feed-forward projection before residual addition.

Each diffusion block applies noise-conditioned Mamba mixing followed by an MLP refinement with residual connections, and the resulting hidden states are propagated through a stack of such blocks to predict token logits at each denoising step. The model defines a categorical distribution over clean tokens and is trained using the standard masked diffusion objective, preserving probabilistic consistency with absorbing-state discrete diffusion. We defer the full architectural formulation, conditioning mechanism, and training details to Appendix A.1. By replacing attention with bidirectional state-space dynamics, `DiffuMamba` achieves efficient and expressive denoising with linear scaling in both sequence length and memory.

Recent results in large-scale autoregressive modeling show that hybrid architectures (Lieber et al., 2024; Bae et al., 2025; Zuo et al., 2025; Wang et al., 2025a) combining attention and state-space layers can outperform both pure Transformer and pure Mamba models under similar compute budgets, leveraging complementary strengths in local recurrence and global dependencies. Motivated by these findings, we investigate hybrid DLMs by interleaving bidirectional Mamba and Transformer blocks, inserting one attention mixer every $N$ mixer blocks[1] (with $N = 5$ in all experiments, following Nemotron-H (Blakeman et al., 2025); an ablation across $N \in \{3, 5, 9, 15\}$ shows every hybrid outperforms both pure endpoints on MAUVE with $N=5$ being a robust default as seen in Appendix B). Extending this paradigm from AR to diffusion, we examine whether similar complementarities hold under masked denoising objectives.

## 4. Experiments

We evaluate three architectures under an identical diffusion objective, masking process, and noise schedule to ensure a controlled comparison. The first configuration, `DiffuTran`, employs an MHA-based mixer in every diffusion block; the second, `DiffuMamba`, replaces all such mixers with bidirectional Mamba blocks; and the third, `DiffuMamba-H`, adopts a hybrid design that interleaves MHA and Mamba blocks. Section 3.2 details the corresponding design principles. Across all variants, the overall architecture, data flow, and conditioning mechanisms remain identical, with only the internal mixer type varying, as illustrated in Figure 1. To maintain comparable parameter budgets, the MLP expansion ratio in `DiffuMamba` and hybrid configurations is set to half that of the MHA-only model, while other hyperparameters, such as the number of blocks, hidden size, etc are kept constant as detailed in Table 7. The residual parameter overhead in `DiffuMamba`/`DiffuMamba-H` does not translate into

---

[1] $N$ is a hyperparameter that can be tuned to balance model performance and inference throughput.

*Table 1.* Zeroshot Perplexities (PPL ↓) across Benchmarks for Different Model Sizes and Configurations. Best PPL are highlighted in blue and second best are underlined. We observe that **DiffuMamba-H** (1.3B) achieves the best overall performance across all configurations, with **DiffuMamba** as the next strongest model. At smaller scales (240M), however, **DiffuMamba** delivers performance comparable to its attention-based diffusion counterpart, indicating that the advantages of hybridization become more pronounced at larger model sizes.

| Model Size | Configuration | PTB | WikiText | LM1B | Lambada | AG News | PubMed | ArXiv | Avg. |
|---|---|---|---|---|---|---|---|---|---|
| | DiffuTran | 153.47 | 49.06 | 142.25 | 49.06 | 105.06 | 29.83 | 43.79 | 81.79 |
| 240M | DiffuMamba | 99.04 | 51.35 | 169.92 | 46.66 | 112.59 | 33.98 | 33.20 | **78.11** |
| | DiffuMamba-H | 147.97 | 50.96 | 145.25 | 46.43 | 114.20 | 31.94 | 30.99 | 81.11 |
| | DiffuTran | 112.69 | 40.76 | 103.51 | 39.73 | 81.49 | 25.51 | 25.37 | 61.29 |
| 0.5B | DiffuMamba | 116.29 | 42.56 | 117.42 | 40.31 | 90.10 | 27.60 | 27.79 | 66.01 |
| | DiffuMamba-H | 110.42 | 40.25 | 100.59 | 39.30 | 78.38 | 26.12 | 25.84 | **60.13** |
| | DiffuTran | 101.14 | 36.63 | 99.72 | 36.73 | 67.53 | 23.03 | 23.25 | 55.43 |
| 1.3B | DiffuMamba | 99.04 | 34.74 | 102.01 | 36.04 | 68.75 | 22.35 | 22.98 | 55.13 |
| | DiffuMamba-H | 96.51 | 31.92 | 92.83 | 34.04 | 63.22 | 19.05 | 20.67 | **51.18** |

more compute: at $L$=8K, `DiffuMamba` uses only $0.82\times$ `DiffuTran`'s per-token FLOPs (Appendix C, Table 10) despite carrying $1.30\times$ the parameters, thus giving a fair, architecture-isolated evaluation of modeling quality, efficiency, and inference throughput. Our goals are to answer:

1. **Performance & Scaling:** Can Mamba-based DLMs reach or even surpass the performance of standard Transformer DLMs at different parameter scales and training tokens scales?

2. **Throughput and Latency:** Does an SSM backbone yield higher tokens/sec and lower per-step latency with identical inference settings?

### 4.1. Data, Tokenization, and Pretraining Setup

We evaluate `DiffuMamba` and `DiffuMamba-H` models as generative LMs by comparing their performance against `DiffuTran` at three parameter scales: 240M, 0.5B, and 1.3B. We further compare throughput and wall-clock model latency at the largest 1.3B scale. All models are trained on the DCLM dataset (Li et al., 2024) using the GPT2 tokenizer (Radford et al., 2019), with a fixed context length of 1,024 tokens. We train all models with a fixed log-linear noise schedule and the masked diffusion objective, under approximate Quokka optimal compute budget (Ni et al., 2025b) as given in the Appendix, Table 7 and Table 8. For measuring inference throughput, we record the wall-clock model latency and tokens-per-second throughput on a single NVIDIA H100 GPU using bf16 precision and PyTorch backend using CUDA-graphs. Each measurement averages over 5 full diffusion decoding runs following 3 warm-up iterations.

### 4.2. Modeling Quality

We first measure validation perplexity (Val. PPL) for different configuration of models as shown in Table 2. Across

*Table 2.* Validation Perplexity (Val. PPL ↓) under Chinchilla (Hoffmann et al., 2022) and Quokka (Ni et al., 2025b) Compute Budgets. Best PPL are highlighted in blue and second best are underlined. For the 1.3B model, `DiffuTran` reaches 25.01 and 22.72 Val. PPL. `DiffuMamba-H` improves by **2** points and yields a **4.3×** inference speedup (see Figure 2); at smaller scales, `DiffuMamba` and `DiffuMamba-H` remain competitive.

| Model | 240M | | 0.5B | | 1.3B | |
|---|---|---|---|---|---|---|
| | Chinchilla | Quokka | Chinchilla | Quokka | Chinchilla | Quokka |
| DiffuTran | 43.82 | 32.11 | 31.03 | 25.07 | 25.01 | 22.72 |
| DiffuMamba | 44.09 | 33.01 | 30.78 | 25.46 | 23.96 | 21.41 |
| DiffuMamba-H | 42.67 | 32.49 | 29.06 | 25.14 | 22.89 | 20.17 |

1.3B parameter scale, hybrid `DiffuMamba-H` consistently outperform `DiffuTran` the pure-attention counterpart under both Chinchilla and Quokka compute budgets. At the 1.3B scale, `DiffuTran` attains 25.01 and 22.72 PPL, but `DiffuMamba-H` achieves the best results with 22.89 and 20.17, delivering roughly a 2 point perplexity reduction. At smaller scales (240M and 0.5B), `DiffuMamba-H` remain competitive, often securing the best or second-best perplexities, demonstrating that hybridization provides consistent gains across pre-training regimes.

We next assess models' ability to generalize zero-shot to held-out datasets. The results are presented in Table 1. Following Sahoo et al. (2024), we evaluate PPL upper bounds on Penn Treebank (PTB; Marcus et al., 1993), WikiText (Merity et al., 2017), LM1B (Chelba et al., 2014), Lambada (Paperno et al., 2016), AG News (Zhang et al., 2015), and Pubmed/Arxiv subsets of Scientific Papers (Cohan et al., 2018). All models are evaluated without fine-tuning.

At the 240M scale, `DiffuTran` outperforms both `DiffuMamba` and `DiffuMamba-H` on 4 out of 7 datasets, indicating that at smaller scales, Mamba-based models struggle to generalize effectively. This trend can also be observed in Table 2, where at small scale `DiffuTran` has lower Val. PPL. At 0.5B, the benefits of hybridization become immediately evident: `DiffuMamba-H` outperforms all other

*Table 3.* Asymptotic inference efficiency at batch size $B=1$. We report FLOPs ($F$), memory operations ($M$), arithmetic intensity (AI= $F/M$), and throughput ($T$). $L$ is sequence length, $d$ hidden dimension, $K$ diffusion steps, $A$ Mamba state size, $G$ block size, and $p$ the step scaling factor. $^{\dagger}$Cost includes block diffusion steps and cache recomputation.

| Model | FLOPs ($F$) | Memory ($M$) | AI ($F/M$) | Throughput ($T$) |
|---|---|---|---|---|
| AR | $\mathcal{O}(Ld^2 + L^2 d)$ | $\mathcal{O}(Ld^2 + L^2 d)$ | $\mathcal{O}(1)$ | $\mathcal{O}\left(\frac{1}{Ld + d^2}\right)$ |
| DiffuTran | $\mathcal{O}(KLd^2 + KL^2 d)$ | $\mathcal{O}(KLd + Kd^2)$ | $\mathcal{O}(L)$ | $\mathcal{O}\left(\frac{C_{\max}}{KLd + Kd^2}\right)$ |
| DiffuMamba | $\mathcal{O}(KLd^2 + KLdA)$ | $\mathcal{O}(KLd + Kd^2 + KLA)$ | $\mathcal{O}\left(\frac{dL}{d+L}\right)$ | $\mathcal{O}\left(\frac{L}{KLd + Kd^2}\right)$ |
| DiffuMamba-H | $\mathcal{O}(KLd^2 + KL^2 d)$ | $\mathcal{O}(KLd + Kd^2)$ | $\mathcal{O}(L)$ | $\mathcal{O}\left(\frac{C_{\max}}{KLd + Kd^2}\right)$ |
| DiffuMamba-H + Fast-dLLM | $\mathcal{O}\left(KGd^2 + KGLd + \frac{L}{G}(Ld^2 + L^2 d)\right)^{\dagger}$ | $\mathcal{O}(KLd + Kd^2)$ | $\mathcal{O}\left(\frac{pL}{G}\right)$ | $\mathcal{O}\left(\frac{C_{\max}}{KLd + Kd^2}\right)$ |

variants on 5 out of 7 datasets, establishing a clear performance lead. The advantage becomes even more pronounced at 1.3B, where DiffuMamba-H surpasses DiffuTran across all datasets, with DiffuMamba ranking second in most cases (5 out of 7) indicating that Mamba's sequence-modeling inductive bias scales more effectively than pure attention for diffusion denoising.

These results highlight a key trend: while Mamba enables efficient long-range token mixing, adding interleaved attention layers in DiffuMamba-H captures complementary global dependencies consistent with findings in AR linear attention (Wang et al., 2025a). This hybrid design consistently improves generalization at larger scales.

*Table 4.* Zero-shot downstream accuracy ($\uparrow$) for 1.3B models. At this scale, overall performance is modest, as 1.3B models struggle on these challenging benchmarks. Both DiffuMamba and DiffuMamba-H consistently outperform DiffuTran, with DiffuMamba-H achieving the strongest results. Best results are shown in blue ; second-best are underlined.

| Model (1.3B) | OBQA | HS | PIQA | LOQA | ARC | Avg. |
|---|---|---|---|---|---|---|
| DiffuTran | 26.61 | 34.97 | 60.64 | 21.86 | 24.98 | 33.81 |
| DiffuMamba | 32.12 | 37.74 | 62.56 | 29.87 | 28.33 | 38.12 |
| DiffuMamba-H | 33.87 | 38.02 | 59.13 | 32.35 | 27.83 | 38.24 |

We also report downstream eval results on OpenBookQA (OBQA (Mihaylov et al., 2018)), HellaSwag (HS (Zellers et al., 2019)), PIQA (Bisk et al., 2020), LogiQA (LOQA (Liu et al., 2020)) and ARC (Clark et al., 2018) for 1.3B models in Table 4. As expected, the absolute scores remain modest at this scale, reflecting the difficulty of these reasoning and knowledge-intensive tasks for base models at 1.3B scale. Nonetheless, a consistent trend emerges across all tasks: DiffuMamba and DiffuMamba-H clearly outperform DiffuTran by $\approx$ **4%** on average, indicating that linear-time state space modeling provides a stronger denoising backbone for diffusion-based LMs. Further hybridizing with attention layers, as in DiffuMamba-H produces the strongest results, suggesting that a small degree of explicit cross-token interaction complements the Mamba backbone. Even at low model capacity where downstream metrics

are weak, Mamba based DLM clearly outperform attention based DiffuTran, demonstrating the effectiveness of state space driven denoising and its superior inference throughput.

**Length generalization and long-range retrieval.** To assess whether DiffuMamba/DiffuMamba-H also generalize to unseen sequence lengths, we additionally evaluate two generation-oriented metrics across three sequence lengths. (i) **Gen PPL** is the perplexity assigned to model-generated samples by an external GPT-2 Large judge (Radford et al., 2019), a standard fluency proxy in prior diffusion-LM work (Sahoo et al., 2024; Nie et al., 2026). (ii) **MAUVE** (Pillutla et al., 2021) is a distributional similarity score between the model's generation distribution and the human reference distribution. The 1.3B models, all trained at $L=1024$, are evaluated at $L \in \{512, 1024, 2048\}$, where $L=2048$ probes *length generalization* to double the training context.

As seen in Table 5, at training length, DiffuMamba-H leads DiffuTran on every metric: Avg. PPL (measured over WikiText, Arxiv and Lambda (Appendix E)) 28.88 vs. 32.20, Gen PPL 37.34 vs. 43.06, MAUVE 0.744 vs. 0.712. At $L=2048$, DiffuTran's zero-shot Avg. PPL severely degrades from 32.20 to 56.57 ($+75.7\%$), while DiffuMamba and DiffuMamba-H are essentially unchanged ($-6.6\%$ and $-1.2\%$) and both see an improvement in Gen PPL by $\approx 47\%$ and in MAUVE by $\approx 7\%$. The Mamba-2 recurrence is position-agnostic and needs no extrapolation, whereas DiffuTran has to extend its position encoding past the training horizon. To further test model's long-range capability, we evaluate Needle-in-a-Haystack which tells the same story: *at $4\times$ training length, DiffuMamba-H retains $28\%$ accuracy vs. DiffuTran's $10\%$* (Appendix F).

### 4.3. Inference Throughput

Figures 2a and 2b compare inference throughput between the attention-based diffusion model, DiffuTran and our proposed DiffuMamba and DiffuMamba-H as se-

*Table 5.* Length generalization at the 1.3B scale, comparing in-distribution ($L$=1024, training length) vs. extrapolation ($L$=2048, double training length). `DiffuTran`'s average zero-shot PPL degrades by +75.7% at $L$=2048 while `DiffuMamba` and `DiffuMamba-H` are essentially stable ($-6.6$% and $-1.2$%). All three models see Gen PPL drop and MAUVE rise at the longer length (more context → more fluent and human-like generations), and `DiffuMamba-H` retains the lead on every metric at every length. $L$=512 results and per-dataset breakdowns are in Appendix E.

| Model (1.3B) | Avg. PPL ↓ | | | Gen PPL ↓ | | | MAUVE ↑ | | |
|---|---|---|---|---|---|---|---|---|---|
| | $L$=1024 | $L$=2048 | $\Delta$ | $L$=1024 | $L$=2048 | $\Delta$ | $L$=1024 | $L$=2048 | $\Delta$ |
| DiffuTran | 32.20 | 56.57 | +75.7% | 43.06 | 25.91 | −39.8% | 0.712 | 0.734 | +3.1% |
| DiffuMamba | 31.25 | 29.18 | −6.6% | 42.06 | 22.54 | −46.4% | 0.704 | 0.758 | +7.7% |
| DiffuMamba-H | **28.88** | **28.53** | −1.2% | **37.34** | **19.43** | −48.0% | **0.744** | **0.796** | +7.0% |

quence length ($L$) increases from 64 to 65536.

> **Takeaway.** From 240M to 1.3B models `DiffuMamba` match or improve upon `DiffuTran` in validation and zero-shot PPL, with `DiffuMamba-H` yielding the most consistent gains at larger scale. `DiffuMamba-H` also wins on Gen PPL and MAUVE, extrapolating cleanly upto 2x the training context whereas `DiffuTran` **degrades** by 76% in Avg. PPL.

Following Nie et al. (2026); Wu et al. (2026), we fix batch size $B = 1$. We are interested in evaluating inference efficiency under different decoding paradigms, hence we include `Fast-dLLM` (Wu et al., 2026) – a training free block diffusion decoding paradigm that relies on KV-cache for inter-block diffusion and *recomputes* KV-cache after each generation block. Throughput ($T$) is measured in tokens per second as the number of generated tokens divided by total wall-clock decoding time. Unlike prior works that use a fixed number of denoising steps (e.g., $K = 128$ in Nie et al. (2026)), we scale the number of steps with the sequence length as $K = L/p$ to enable long-context evaluation, and report results for $p \in \{8, 16\}$. For `Fast-dLLM` we fix the block size to $G = 32$ following Wu et al. (2026), yielding denoising steps per block, $k = 32/p$.

It can be observed that for moderate sequence lengths ($L \leq 2K$), `DiffuTran+Fast-dLLM` achieves the highest throughput among all models. In this regime, `DiffuMamba` and `DiffuTran` exhibit competitive and nearly identical performance. As the sequence length increases, throughput gradually decreases for all methods, consistent with the asymptotic analysis in Table 3. In particular, throughput becomes dominated by the hidden dimension $d$ rather than by $L$, staying approximately constant in this regime.

Beyond $2K$ tokens in Figures 2a and 2b, the throughput of `DiffuTran` degrades sharply, while `DiffuMamba` (memory bandwidth bound (Baruah et al., 2025)) and `DiffuMamba-H` (reduces FLOPs per forward) experience a substantially slower decline. This trend aligns with the asymptotic throughput analysis in Table 3, which predicts more favorable scaling behavior for `DiffuMamba` and

`DiffuMamba-H`. When $L > d$ (1920) (approximately at $L = 2048$), the diffusion model enters a compute-saturated regime. In this regime, the FLOPs per token grow with $L$, while the sustained FLOPs/s remain bounded by the peak hardware capacity $C_{\max}$, resulting in a decline in throughput. Beyond this point, `DiffuTran` exhibits the steepest throughput collapse due to the combined effects of quadratic attention cost and linearly increasing denoising steps, resulting in $T = \mathcal{O}(\frac{C_{max}}{KLd+Kd^2}) \approx \mathcal{O}(\frac{1}{L^2})$, where $K = L/p$, scaling at long context lengths. In contrast, `DiffuMamba` is memory bound. As a result, for $L > d$, throughput scales as $\mathcal{O}(\frac{L}{Kd^2+KLd})$, where $K = L/p$, thus $T = \mathcal{O}(\frac{p}{d^2+Ld}) = \mathcal{O}(\frac{1}{L})$. This gives slower throughput degradation, retaining a $\mathbf{8.2\times}$ and $\mathbf{4.3\times}$ advantage at $K = L/16$ denoising steps for `DiffuMamba` and `DiffuMamba-H` respectively over `DiffuTran` at 65K tokens.

In Figures 2c and 2d, we evaluate all DLMs using a block-autoregressive inference that reuses cached representations across successive generation blocks (Wang et al., 2025b; Arriola et al., 2025). Concretely, once a block of tokens is denoised, the corresponding cache is retained and directly reused when processing the next block, avoiding repeated forward passes over previously generated tokens. As generation proceeds, caches accumulate incrementally across blocks, enabling efficient long-context inference.

*Table 6.* Block-cache generation quality (Gen PPL ↓, GPT-2 Large judge, 128 samples per cell). At long sequence $L$=16K with $G$=128, `DiffuMamba` achieves the best Gen PPL (**12.09**).

| Model (1.3B) | $L = 8K$ | | | $L = 16K$ | | |
|---|---|---|---|---|---|---|
| | $G$=32 | $G$=64 | $G$=128 | $G$=32 | $G$=64 | $G$=128 |
| DiffuTran | 19.60 | 19.81 | 22.20 | **15.65** | 17.34 | 22.61 |
| DiffuMamba | 52.77 | 18.53 | **12.84** | 43.62 | 20.42 | **12.09** |
| DiffuMamba-H | **17.67** | **14.63** | 33.54 | 18.76 | **14.58** | 31.24 |

Under this setting, all DLMs recover a clear advantage over autoregressive baselines. Further, at long generation length ($L = 260K$), `DiffuMamba` achieves a $\mathbf{2.3\times}$ throughput improvement and `DiffuMamba-H` a $\mathbf{1.9\times}$ improvement over `DiffuTran`. This inference paradigm is particularly well suited to Mamba-based diffusion: *state updates can be constructed locally within each block and reused across*

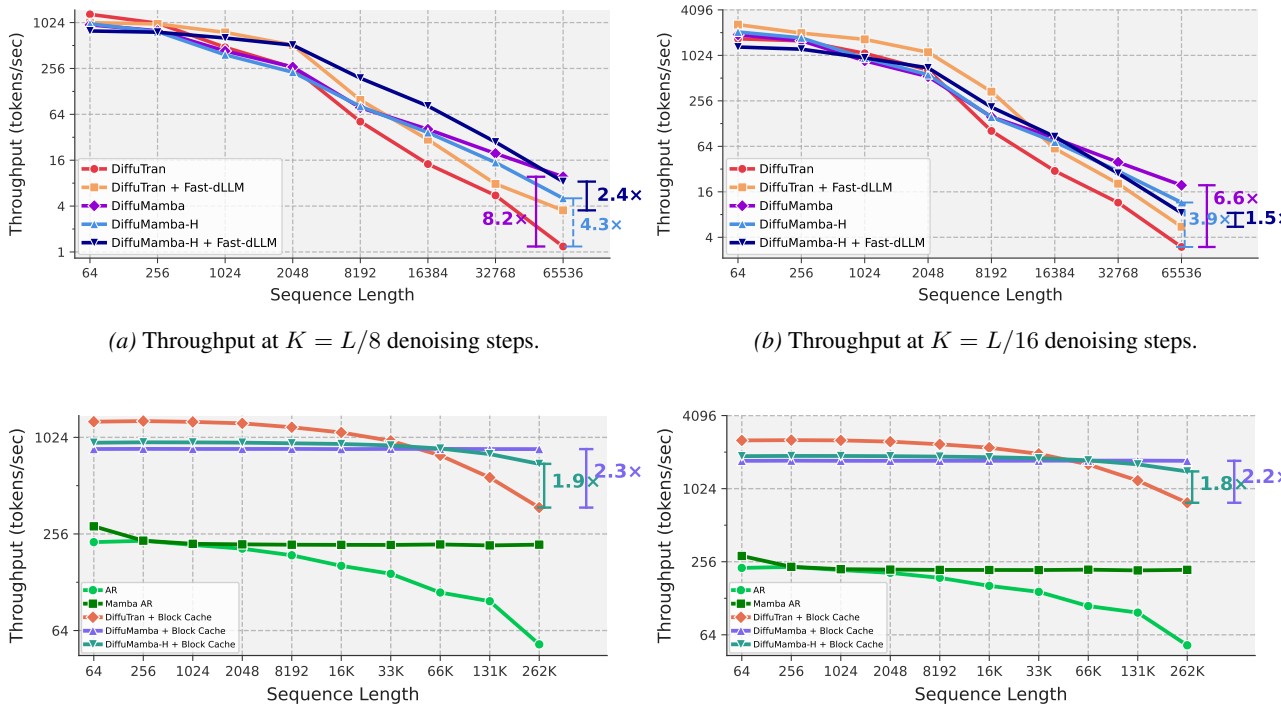

*(a)* Throughput at $K = L/8$ denoising steps.

*(b)* Throughput at $K = L/16$ denoising steps.

*(c)* Throughput at $K = L/8$ denoising steps.

*(d)* Throughput at $K = L/16$ denoising steps.

*Figure 2.* Inference throughput for 1.3B models vs. sequence length ($L$) with a batch size of 1 and $L/p$ decoding steps where $p \in \{8, 16\}$. **(a),(b)** are *compute bound baselines* while **(c),(d)** are the *memory bound ones*. **(a),(b)**: `DiffuMamba` and `DiffuMamba-H` yield **8.2×** and **4.3×** throughput improvement over `DiffuTran` respectively. Further `DiffuMamba-H + Fast-dLLM` (block size = 32) boosts throughput at long sequences and gives **2.4×** improvement over `DiffuTran + Fast-dLLM` (Wu et al., 2026). **(c),(d)** `DiffuMamba` and `DiffuMamba-H` achieve **2.3×** and **1.9×** throughput improvements over `DiffuTran`, respectively, when equipped with simple block caching similar to Wang et al. (2025b). Further, block caching delivers significantly higher throughput than autoregressive baselines for both Attention and Mamba-based models. Unlike Fast-dLLM, where the cache is recomputed after each block, our approach enables block-wise autoregressive generation to enable cache utilization.

*blocks in an autoregressive fashion.* In our bidirectional Mamba design, right-to-left state updates are restricted to operate only within the current block, while cached states are propagated forward across blocks. These results highlight that eliminating cache recomputation is key to outperforming autoregressive baselines at long contexts.

Restricting the right-to-left Mamba state to within each block (rather than the full sequence) does not significantly hurt generation quality: `DiffuMamba` in fact achieves its best Gen PPL (12.09 at $L$=16K, $G$=128) at the same block size that maximises throughput, and `DiffuMamba-H` peaks at $G$=64 (Table 6). The throughput gains also hold at larger batch: at $B$=8 and $L$=16K, `DiffuMamba` retains a **3.5×** advantage and `DiffuMamba-H` a **2.0×** advantage over `DiffuTran`, with `DiffuMamba-H` + Fast-dLLM the overall leader (Appendix D).

To further understand the throughput trends in Figure 2, we analyze the *per-forward-pass latency* as a function of sequence length $L$. From each benchmark record, we compute the mean latency per denoising step as $t(L) =$ $\frac{\text{Model Total Time}}{\text{steps}}$ which isolates the cost of a single model forward pass at sequence length $L$. We model the measured latency using a non-negative quadratic decomposition, $t(L) = aL^2 + bL + c$ s.t. $a, b, c \geq 0$, where the quadratic term $aL^2$ captures attention-like pairwise token interactions, the linear term $bL$ captures token-wise computations (e.g., MLP/SSM/projections), and the constant term $c$ captures sequence-length–independent overheads.

The fitted equations make the scaling behavior explicit as demonstrated in Figure 3. Attention-based diffusion (`DiffuTran`) has a comparatively larger quadratic coefficient, so the quadratic component $aL^2$ becomes dominant as $L$ grows. In contrast, Mamba-based diffusion (`DiffuMamba`) exhibits a much smaller quadratic coefficient; over the evaluated range, latency is largely explained by the linear and constant components. Hybrid interleavings (`DiffuMamba-H` variants) sit between these regimes, retaining a controlled quadratic contribution due to periodic attention insertion.

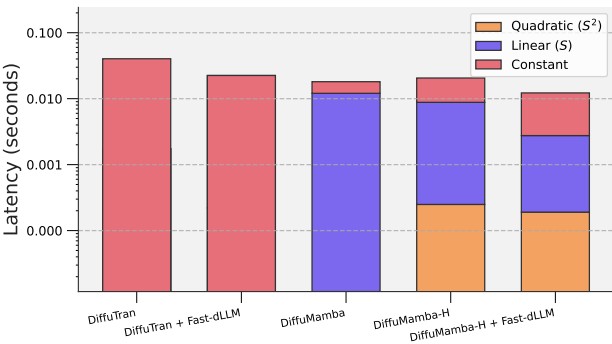

*(a)* Latency Components at $L = 1024$

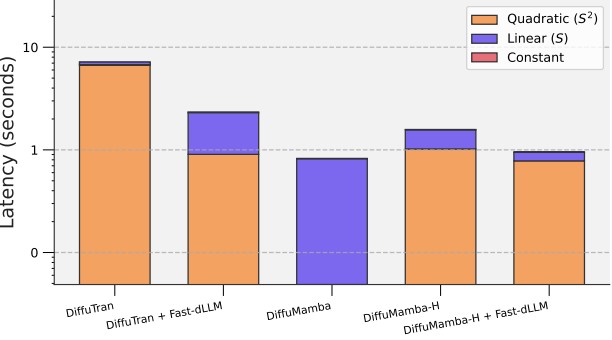

*(b)* Latency Components at $L = 65K$

*Figure 3.* Latency decomposition per denoising step for a single forward pass across different model configurations, obtained by fitting a non-negative quadratic model $t(L) = aL^2 + bL + c$ to the measured latency curves. The quadratic coefficient $a$ captures attention-like pairwise interactions, while the linear coefficient $b$ reflects token-wise computations (e.g., MLP/SSM), and $c$ denotes constant overheads. For intermediate sequence lengths, linear and constant terms dominate across models. At large lengths, the superior scaling of `DiffuMamba` stems from its negligible quadratic component, whereas `DiffuTran` becomes dominated by the quadratic term. Although `DiffuMamba-H` reduces latency relative to `DiffuTran`, its scaling remains governed by attention.

**Takeaway** Mamba-based DLMs achieve higher throughput than Transformer based ones under all inference algorithms. Block caching removes quadratic re-computation in Transformers pushing DLMs' throughput far beyond AR. Mamba-based block-AR DLMs with constant memory requirement achieve best throughput in this memory-bound setting. With Block Caching `DiffuMamba` achieves its best Gen PPL (12.09) at the same $G{=}128$ that maximizes throughput (Table 6). These conclusions are robust to batch size: at $B{=}8$, `DiffuMamba` retains a $3.5\times$ throughput advantage over `DiffuTran` at $L{=}16$K (Appendix D).

Taken together, this latency decomposition clarifies the fundamental source of the throughput advantages observed in Figure 2. By isolating the role of quadratic attention costs, our analysis explains why Mamba-based diffusion exhibits more favorable scaling and why hybrid designs interpolate smoothly between the two regimes. These insights motivate the architectural choices we summarize next and provide a principled foundation for the conclusions that follow.

## 5. Conclusion

This work demonstrates the feasibility of DLMs built with Mamba backbones. We introduce `DiffuMamba`, the first diffusion LM that relies exclusively on linear state-space mixers instead of multi-head attention, and `DiffuMamba-H`, a hybrid variant that interleaves Mamba-2 and attention layers to capture complementary global context. Across model scales from 240M to 1.3B parameters, both approaches match or surpass the Transformer-based baseline `DiffuTran` in language modeling performance.

Our throughput analysis demonstrates that Mamba-based DLMs scale more favorably than MHA-based DLMs across a range of inference algorithms. In particular, DLMs relying on block cache reuse yield the strongest performance across the entire context length spectrum efficiently combining the multi-token generation advantages of DLMs and preventing quadratic scaling by fixing the representations of the past tokens (e.g. using KV-cache). While `DiffuMamba + Block Cache` emerges as the most efficient design for long sequence generation, achieving optimal performance in this regime requires training with block-diffusion inductive biases or incorporating distillation objectives as in Wang et al. (2025b). As the primary goal of this work is to establish feasibility and characterize the efficiency advantages of DLMs with linear mixers, we leave the training of block-cached hybrid models at useful scales and the exploration of alternative linear mixers to future work. We summarize the main limitations of this work in Appendix G.

## Impact Statement

This paper presents work whose goal is to advance the field of Machine Learning. There are many potential societal consequences of our work, none which we feel must be specifically highlighted here.

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

# A. Model Configurations

Model configurations for `DiffuTran`, `DiffuMamba` and `DiffuMamba-H` in our experiments. All variants are trained under the same diffusion objective, noise schedule, and tokenization. The only difference lies in the internal mixer architecture. The MLP expansion ratio in `DiffuMamba` and `DiffuMamba-H` configurations is reduced by half for comparable parameter budgets. In the hybrid architecture, an attention layer is inserted every $N$ mixer layers; we set $N = 5$ in all experiments. In Table 8, we list down all the training and inference hyperparameters.

*Table 7.* Model configurations for `DiffuTran`, `DiffuMamba` and `DiffuMamba-H`.

| Model Size | Configuration | Layers | $d_{model}$ | $d_{mlp}$ | $d_{head}$ | $d_{state}$ | Context Len. | # Tokens |
|---|---|---|---|---|---|---|---|---|
| | DiffuTran | 24 | 960 | 960*4 | 32 | - | 1024 | 25B |
| 240M | DiffuMamba | 24 | 960 | 960*2 | 32 | 128 | 1024 | 25B |
| | DiffuMamba-H (N=5) | 24 | 960 | 960*2 | 32 | 128 | 1024 | 25B |
| | DiffuTran | 36 | 960 | 960*4 | 32 | - | 1024 | 50B |
| 0.5B | DiffuMamba | 36 | 960 | 960*2 | 32 | 128 | 1024 | 50B |
| | DiffuMamba-H (N=5) | 36 | 960 | 960*2 | 32 | 128 | 1024 | 50B |
| | DiffuTran | 24 | 1920 | 1920*4 | 32 | - | 1024 | 120B |
| 1.3B | DiffuMamba | 24 | 1920 | 1920*2 | 32 | 128 | 1024 | 120B |
| | DiffuMamba-H (N=5) | 24 | 1920 | 1920*2 | 32 | 128 | 1024 | 120B |

## A.1. Architecture Details of DiffuMamba

We present here the architectural details of `DiffuMamba` an MDM whose denoiser is built on a *bidirectional state-space Mamba (BiMamba)* backbone. DiffuMamba preserves the probabilistic structure of masked discrete diffusion while replacing the Transformer encoder with a bidirectional Mamba mixer, enabling *linear-time* inference and substantially lower memory overhead during multi-step denoising.

Let $\mathbf{x} \in \mathbb{R}^{B \times L \times d}$ denote a batch of token embeddings, where $B$ is the batch size, $L$ the sequence length, and $d$ the hidden dimension. Each block is composed of two independent Mamba layers: one processes the sequence in the forward direction, and the other processes the sequence in the reverse direction

$$\mathbf{h}_i^{\rightarrow} = A_f * \mathbf{h}_{i-1}^{\rightarrow} + B_f * \mathbf{x}_i, \quad i = 1, \dots, L, \tag{7}$$

$$\mathbf{h}_i^{\leftarrow} = A_b * \mathbf{h}_{i+1}^{\leftarrow} + B_b * \mathbf{x}_i, \quad i = L, \dots, 1. \tag{8}$$

Here $A_f, B_f, A_b, B_b \in \mathbb{R}^{k \times d}$ are learnable state-transition kernels implemented as 1D causal and anti-causal convolutions or efficient scan operations (Gu & Dao, 2024). The two directional feature streams are fused through simple *additive integration*

$$\text{Mamba}(x_i) = \mathbf{h}_i = \mathbf{h}_i^{\rightarrow} + \mathbf{h}_i^{\leftarrow}, \tag{9}$$

providing a symmetric context representation while maintaining numerical stability. The resulting hidden sequence $\mathbf{H} = (\mathbf{h}_1, \dots, \mathbf{h}_L)$ is then normalized and passed through a lightweight feed-forward projection before residual addition.

Each Diffusion Block employs timestep-conditioned adaptive layer normalization (AdaLN) to inject the diffusion noise level into the hidden activations. A small MLP maps the scalar timestep $t$ to a continuous embedding $\tau_t = \text{MLP}(t) \in \mathbb{R}^{d_c}$, which modulates both the mixer and MLP sublayers:

$$\text{AdaLN}(\mathbf{x}; \tau_t) = \gamma_t \cdot \frac{\mathbf{x} - \mu(\mathbf{x})}{\sigma(\mathbf{x})} + \beta_t, \quad (\gamma_t, \beta_t) = W_{\text{cond}} \tau_t. \tag{10}$$

This conditioning allows the denoiser to adapt its internal recurrence dynamics to varying noise levels across diffusion steps.

Each block then applies Mamba mixing followed by an MLP refinement as shown in Figure 1:

$$\mathbf{y}_{\text{mixer}} = \text{Mamba}(\text{AdaLN}(\mathbf{x}; \tau_t)) + \mathbf{x}, \tag{11}$$

$$\mathbf{y}_{\text{mlp}} = \text{MLP}(\text{AdaLN}(\mathbf{y}_{\text{mixer}}; \tau_t)) + \mathbf{y}_{\text{mixer}}. \tag{12}$$

The final output $\mathbf{y}_{\text{mlp}}$ is passed to the next block (B) in the stack. During denoising, given masked input embeddings $\mathbf{x}_t$ and timestep embedding $\tau_t$, the model predicts token logits via the full diffusion stack:

$$\mathbf{z} = f_\theta(\mathbf{x}_t, \tau_t) = \text{OutputProj}(B_N(\cdots B_1(E(\mathbf{x}_t), \tau_t)\cdots)). \tag{13}$$

The conditional distribution over clean tokens is

$$p_\theta(\mathbf{x}_0|\mathbf{x}_t, t) = \prod_{i=1}^{L} \text{Cat}(x_0^i; \text{softmax}(z_i)). \tag{14}$$

Training minimizes the masked diffusion objective (Eq. 2), preserving probabilistic consistency with absorbing-state discrete diffusion.

*Table 8.* Key training and inference hyperparameters used across all experiments unless stated otherwise.

| Category | Setting |
| --- | --- |
| Backbone | Transformer (`DiffuTran`), Mamba (`DiffuMamba`), Hybrid (`DiffuMamba-H`) |
| Diffusion Type | Absorbing-state masked diffusion |
| Parameterization | Substitution (subs) |
| Noise Conditioning | Log Linear |
| Global Batch Size | 512 |
| Precision | bf16 |
| EMA Decay | 0.9999 |
| Antithetic Sampling | Enabled |
| Sampling $\epsilon$ | $10^{-3}$ |
| Gradient Clipping | 1.0 |
| Optimizer | Adam |
| Max Learning Rate | $1 \times 10^{-4}$ |
| Min Learning Rate | $1 \times 10^{-6}$ |
| Weight Decay | 0.1 |
| Adam Betas | (0.9, 0.95) |
| Adam $\epsilon$ | $10^{-8}$ |
| LR scheduler | Cosine |
| Denoising Steps factor (p) | {8, 16} |
| Block Length | 32 |
| Inference Batch Size | 1 |
| hidden dimension | 1920 |
| # blocks | 24 |
| # heads | 24 |
| bidirectional weight tie | False |
| mamba state dimension | 128 |
| mamba conv dimension | 4 |
| expansion factor | 2 |
| interleaving attention (N) | 5 |

# B. Hybrid Interleaving-Rate Ablation

This subsection provides the full ablation justifying the choice $N=5$ for the hybrid `DiffuMamba-H` architecture (referenced from section 3 and subsection 4.2). $N$ is the period of the Attention–Mamba repeating unit (one attention mixer followed by $N-1$ bidirectional Mamba mixers).

We sweep $N \in \{3, 5, 9, 15\}$ together with the two pure endpoints (pure attention and pure Mamba) at the 0.5B scale (36 layers) under identical training budgets (50B tokens, identical hyperparameters), and report unconditional Gen PPL (GPT-2 Large judge) and MAUVE on 512 samples.

*Table 9.* Hybrid interleaving-rate ablation at the 0.5B scale (36 layers, 50B training tokens, same hyperparameters as the main experiments). $N$ is the period of the Attention–Mamba repeating unit (one attention mixer followed by $N-1$ bidirectional Mamba mixers); "Attn %" is the resulting fraction of attention layers in the stack, $1/N$.. Every hybrid configuration outperforms *both* pure endpoints on MAUVE, and Gen PPL varies only between 48.07 and 50.97 (a 6% range) across the entire spectrum from pure Mamba to pure attention. $N=5$ is within 3% of the best Gen PPL and within 1% of the best MAUVE while spending only 7 of 36 layers on attention, a strong default rather than a fragile optimum.

| Configuration | Attn % | Gen PPL ↓ | MAUVE ↑ |
|---|---|---|---|
| Pure attention (`DiffuTran`) | 100% | 50.97 | 0.672 |
| `DiffuMamba-H`, $N=3$ | 33% | 48.87 | 0.721 |
| `DiffuMamba-H`, $N=5$ (default) | 20% | 49.37 | 0.719 |
| `DiffuMamba-H`, $N=9$ | 11% | 50.33 | **0.724** |
| `DiffuMamba-H`, $N=15$ | 7% | 49.56 | 0.681 |
| Pure Mamba (`DiffuMamba`) | 0% | **48.07** | 0.674 |

Three observations follow. (i) *The two mixer types are complementary*: every hybrid configuration outperforms *both* pure endpoints on MAUVE ($\geq 0.681$ for any $N \in \{3, 5, 9, 15\}$ vs. 0.672 for pure attention and 0.674 for pure Mamba), indicating that Mamba and attention contribute distinct properties (fluency and distributional diversity respectively) that combine constructively. (ii) *The quality plateau is flat*: Gen PPL varies only between 48.07 and 50.97 across all six configurations (a 6% relative range), so $N=5$ is a robust default rather than a tightly tuned optimum. (iii) $N=5$ *sits at a strong efficiency–quality tradeoff*: it achieves Gen PPL within 3% of the best configuration ($N=3$) and MAUVE within 1% of the best ($N=9$), while using only 7 of 36 attention layers, which directly translates to the FLOPs and throughput advantages reported in Table 10 and Figure 2.

## C. Per-Token FLOPs Derivation

Table 10 summarises the compute–quality comparison at the 1.3B scale; the remainder of this subsection derives the per-token forward FLOPs reported in that table from the trained checkpoint weight shapes.

*Table 10.* Compute–quality comparison at the 1.3B scale. Despite carrying 30% more parameters than `DiffuTran`, `DiffuMamba` uses $0.82\times$ the per-token FLOPs of `DiffuTran` at $L=8$K because Mamba's extra parameters live in $\mathcal{O}(d^2)$ projections that are constant in $L$, while attention's parameter-free $\mathcal{O}(Ld)$ term ($QK^\top + \text{Attn}\cdot V$) dominates at long contexts. "Train FLOPs/tok" is the standard $3\times$ forward estimate (forward + backward + activation gradient); "Total" is over the 120B training tokens ($1\,\text{ZF} = 10^{21}$ FLOPs). Validation perplexities are reproduced from Table 2 (Quokka). Per-layer derivations follow below.

| Model (1.3B) | Params | Param ratio | FLOPs/tok ($L=1$K) | FLOPs/tok ($L=8$K) | Train FLOPs/tok (fwd $\times3$) | Total (120B) | Val. PPL ↓ |
|---|---|---|---|---|---|---|---|
| `DiffuTran` | 1291M | $1.00\times$ | 2.58G | 3.90G | 7.73G | 0.94 ZF | 22.72 |
| `DiffuMamba` | 1681M | $1.30\times$ | 3.21G | **3.21G** ($0.82\times$) | 9.64G | 1.15 ZF | 21.41 |
| `DiffuMamba-H` | 1526M | $1.18\times$ | 2.93G | **3.21G** ($0.82\times$) | 8.80G | 1.08 ZF | **20.17** |

**Compute-matched interpretation.** Two observations follow. First, the additional parameters in Mamba-2 are concentrated in the in/out projection matrices, each contributing $\mathcal{O}(d^2)$ FLOPs per token *independent of $L$*, exactly the same scaling as the QKV and output projections in attention. Second, attention adds an $\mathcal{O}(Ld)$ term per layer (the $QK^\top$ and $\text{Attn}\cdot V$ matmuls) that has *zero learnable parameters* but consumes a growing share of FLOPs at long contexts ($\approx 83\%$ of `DiffuTran`'s per-token FLOPs at $L=65$K). The FLOPs ranking therefore inverts beyond $L \approx 8$K: at $L=8$K, `DiffuMamba` uses $0.82\times$ the FLOPs of `DiffuTran` despite having $1.30\times$ the parameters. The PPL improvements of `DiffuMamba` and `DiffuMamba-H` over `DiffuTran` at the 1.3B scale therefore cannot be attributed to additional parameters or per-token compute, and instead reflect an architectural advantage of bidirectional state-space mixing. We note further that a parameter-matched comparison would actually disadvantage `DiffuTran` at long contexts, since enlarging `DiffuTran` scales its parameter-free attention term upward and widens, not closes, the FLOP gap.

All values in Table 10 are computed directly from the trained 1.3B-scale checkpoint weight shapes and verified against the closed-form formulas listed in the Formula columns below.

**Counting conventions.** For each linear layer with weight matrix $W \in \mathbb{R}^{m \times n}$, we count $2mn$ forward FLOPs per token (one multiply and one add per output element). The parameter-free attention matmuls $QK^\top$ and $\text{Attn} \cdot V$ each contribute $2Ld$ FLOPs per layer per token, totalling the standard $4Ld$ (Kaplan et al., 2020). For the bidirectional Mamba-2 block, we count the input projection, the SSM scan ($2\,d_{\text{inner}}\,d_{\text{state}}$ per token per direction), the small short Conv1d (conv_ch $\cdot\, d_{\text{conv}}$ per token), and the output projection, then multiply by two for the forward and reverse passes. AdaLN modulation contributes $2 \cdot 6d \cdot d_c$ FLOPs per layer where $d_c$ is the timestep embedding dimension. The input embedding lookup, final RMSNorm, and output unembedding contribute a shared 193.0M FLOPs/token across all three models.

The 1.3B configurations correspond to those in Table 7: $d = 1920$, 24 layers, with mlp expansion ratio 4 in `DiffuTran` and 2 in `DiffuMamba`; the hybrid `DiffuMamba-H` uses 5 attention layers (with the same mlp_ratio= 2 as `DiffuMamba`, for parameter parity) and 19 Mamba-2 layers.

*Table 11.* Per-token forward FLOPs of one Transformer layer in `DiffuTran`-1.3B ($d$=1920, $d_{\text{ff}}$=4d=7680, $d_c$=128). The only $L$-dependent term is the attention matmul $QK^\top + \text{Attn} \cdot V$, which has *zero learnable parameters*. At $L$=1024 this term is 7.86 M FLOPs/layer; at $L$=65K it grows to 503.3 M FLOPs/layer, dominating the layer total.

| Operation | Formula | Scaling | FLOPs at $L$=1024 |
|---|---|---|---|
| QKV projections | $6d^2$ | $\mathcal{O}(d^2)$, $L$-indep. | $6(1920)^2 = 22.12\,\text{M}$ |
| $QK^\top + \text{Attn} \cdot V$ | $4Ld$ | $\mathcal{O}(Ld)$, grows with $L$ | $4(1024)(1920) = 7.86\,\text{M}$ |
| Output projection | $2d^2$ | $\mathcal{O}(d^2)$, $L$-indep. | $2(1920)^2 = 7.37\,\text{M}$ |
| MLP (up + down) | $2d \cdot d_{\text{ff}} + 2d_{\text{ff}} \cdot d$ | $\mathcal{O}(d^2)$, $L$-indep. | $2(1920)(7680) \times 2 = 58.98\,\text{M}$ |
| AdaLN | $2 \cdot 6d \cdot d_c$ | $\mathcal{O}(d)$, $L$-indep. | $2(11520)(128) = 2.95\,\text{M}$ |
| **Layer total** | $8d^2 + 4Ld + 4d \cdot d_{\text{ff}} + 12d \cdot d_c$ | | **99.28 M** |

*Table 12.* Per-token forward FLOPs of one bidirectional Mamba-2 layer in `DiffuMamba`-1.3B ($d$=1920, $d_{\text{inner}}$=2d=3840, $d_{\text{state}}$=128, $d_{\text{conv}}$=4, $d_{\text{ff}}$=2d=3840). $D_{\text{in}}$=8056 is the projected dimension before the SSM (combines input projection, gate, $\Delta$, $B$, $C$, and per-head bias parameters). All terms are constant in $L$, so the layer total is identical at every sequence length.

| Operation | Formula | Scaling | FLOPs at $L$=1024 |
|---|---|---|---|
| in_proj (1 dir) | $2d \cdot D_{\text{in}}$ | $\mathcal{O}(d^2)$, $L$-indep. | $2(1920)(8056) = 30.93\,\text{M}$ |
| Conv1d (1 dir) | conv_ch $\cdot\, d_{\text{conv}}$ | $\mathcal{O}(d)$, negligible | $4096 \times 4 = 0.016\,\text{M}$ |
| SSM scan (1 dir) | $2\,d_{\text{inner}} \cdot d_{\text{state}}$ | $\mathcal{O}(d \cdot d_{\text{state}})$, $L$-indep. | $2(3840)(128) = 0.98\,\text{M}$ |
| out_proj (1 dir) | $2d_{\text{inner}} \cdot d$ | $\mathcal{O}(d^2)$, $L$-indep. | $2(3840)(1920) = 14.75\,\text{M}$ |
| *One-direction subtotal* | | | 46.68 M |
| Bidirectional ($\times 2$) | $2 \times$ (one direction) | $\mathcal{O}(d^2)$, $L$-indep. | $2 \times 46.68 = 93.35\,\text{M}$ |
| MLP (up + down) | $4d \cdot d_{\text{ff}}$ | $\mathcal{O}(d^2)$, $L$-indep. | $4(1920)(3840) = 29.49\,\text{M}$ |
| AdaLN | $2 \cdot 6d \cdot d_c$ | $\mathcal{O}(d)$, $L$-indep. | $2(11520)(128) = 2.95\,\text{M}$ |
| **Layer total** | All terms constant in $L$ | | **125.79 M** |

*Table 13.* Total forward FLOPs per token at the 1.3B scale, $L$=1024. The hybrid `DiffuMamba-H` uses 5 attention layers (with mlp_ratio=2 to match `DiffuMamba`, giving 69.79 M FLOPs/layer instead of `DiffuTran`'s 99.28 M) and 19 Mamba-2 layers. The 193.0 M "shared head" contribution covers the input embedding, output unembedding, and final norm and is identical across all three models.

| Model (1.3B) | Params | Layer composition | FLOPs/token |
|---|---|---|---|
| `DiffuTran` | 1291M | $24 \times 99.28\,\text{M} + 193.0\,\text{M}$ | 2.576 G |
| `DiffuMamba` | 1681M | $24 \times 125.79\,\text{M} + 193.0\,\text{M}$ | 3.212 G |
| `DiffuMamba-H` | 1526M | $5 \times 69.79\,\text{M} + 19 \times 125.79\,\text{M} + 193.0\,\text{M}$ | 2.932 G |

**Implication for the parameter-vs-compute question.** The key implication of Table 14 is the crossover behavior: `DiffuTran`'s per-token cost grows linearly in $L$ from 2.58 G at $L$=1K to 14.47 G at $L$=65K (a 5.6× increase), while `DiffuMamba`'s cost is exactly constant in $L$ (a defining property of state-space sequence mixers) and `DiffuMamba-H`'s cost grows only mildly because of its 5 attention layers. This is the source of `DiffuMamba`'s and `DiffuMamba-H`'s inference-time advantage and explains why a parameter-matched comparison would actually *disadvantage* `DiffuTran` at long contexts: matching parameters by enlarging `DiffuTran` further inflates the parameter-free attention term and worsens,

*Table 14.* Per-token forward FLOPs at varying sequence length (1.3B scale). `DiffuTran`'s $\mathcal{O}(L)$-growing attention term contributes $24 \times 4Ld \approx 189\,M, 1510\,M, 12080\,M$ FLOPs at $L = 1K, 8K, 65K$ respectively, reaching 83% of `DiffuTran`'s per-token FLOPs at $L=65K$. `DiffuMamba` is exactly constant in $L$ (every term in Table 12 is $L$-independent); `DiffuMamba-H` grows only because of its 5 attention layers. The crossover where `DiffuMamba` FLOPs $\leq$ `DiffuTran` FLOPs occurs near $L \approx 8K$.

| Model (1.3B) | Params | $L$=1K | $L$=8K | $L$=65K |
|---|---|---|---|---|
| DiffuTran | 1291M | 2.576 G | 3.897 G | **14.467 G** |
| DiffuMamba | 1681M | 3.212 G | 3.212 G | **3.212 G** |
| DiffuMamba-H | 1526M | 2.932 G | 3.207 G | 5.409 G |

not improves, its FLOP profile at $L \geq 8K$. We conclude that the perplexity gains of `DiffuMamba` and `DiffuMamba-H` over `DiffuTran` in Table 2 are best attributed to the bidirectional state-space mixing inductive bias rather than to parameter or compute differences.

## D. Throughput at Larger Batch Size

This subsection complements the $B$=1 throughput analysis of subsection 4.3 (Figures 2a–2d) with a corresponding sweep at $B$=8. The 1.3B-scale models (`DiffuTran`, `DiffuMamba`, `DiffuMamba-H`), Fast-dLLM block size ($G$=32), and denoising-step schedule ($K$=$L/p$ with $p$=16) are all identical to the main-text setup; only the batch size is changed from 1 to 8. We sweep $L \in \{64, 256, 1024, 2048, 8192, 16K\}$ and report the wall-clock throughput in tokens/sec. At $B$=8, all five model variants OOM beyond $L$=16K on the same single-GPU setup that supports $L \leq 65K$ at $B$=1, so the sweep is shorter at the long-$L$ end.

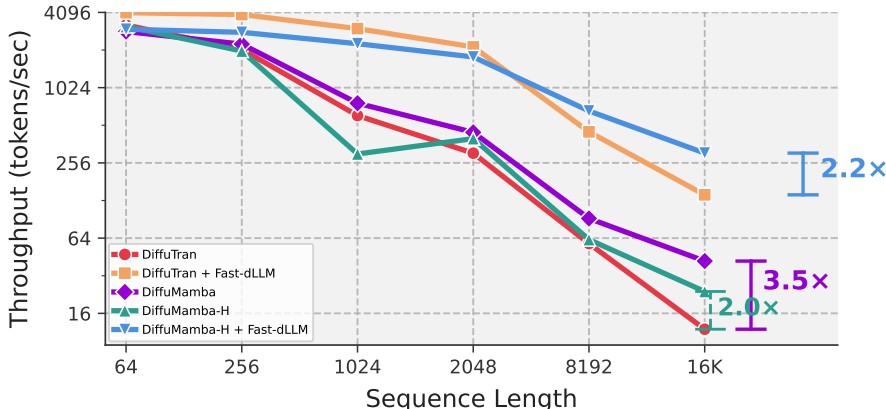

*Figure 4.* Inference throughput at $B$=8 for the 1.3B variants across $L \in \{64..16K\}$ (log–log axes). Highlighted speedups at $L$=16K: `DiffuMamba` achieves $\mathbf{3.5\times}$ throughput over `DiffuTran`, `DiffuMamba-H` achieves $\mathbf{2.0\times}$, and `DiffuMamba-H`+Fast-dLLM achieves $\mathbf{2.2\times}$ over uncached `DiffuMamba-H` (and is the overall throughput leader at this length). At $L \leq 1K$ all five variants are tightly bunched, matching the $B$=1 behavior in Figure 2. We experience OOM at $L > 16K$ at this batch size on the same hardware that supports $L \leq 65K$ at $B$=1.

The qualitative picture at $B$=8 matches the $B$=1 figure (Figure 2). For $L \leq 1K$, all five variants achieve essentially identical throughput ($\sim$3–4K tok/s); the kernels are not yet the bottleneck and the choice of mixer/decoding paradigm makes little difference. As $L$ grows past the 2K hidden-dimension crossover, the curves separate: `DiffuTran`'s throughput collapses fastest (consistent with the $\mathcal{O}(1/L^2)$ compute-bound scaling derived in the main text), `DiffuMamba` and `DiffuMamba-H` degrade more slowly, and Fast-dLLM lifts both `DiffuTran` and `DiffuMamba-H` at long $L$. At the rightmost point $L$=16K, the three speedups quoted in the main text are visible directly on the plot: `DiffuMamba` is $\mathbf{3.5\times}$ `DiffuTran`, `DiffuMamba-H` is $\mathbf{2.0\times}$ `DiffuTran`, and `DiffuMamba-H`+Fast-dLLM is $\mathbf{2.2\times}$ uncached `DiffuMamba-H` and the highest-throughput configuration overall ($\approx$ 256 tok/s). Together with the $B$=1 results in the main text, this confirms that the architectural throughput advantages of Mamba-based diffusion are not an artifact of single-batch evaluation.

## E. Length Generalization: Per-Length Results and Sample Outputs

This subsection provides the full per-length results that underpin Table 5 of the main text. We evaluate the three 1.3B-scale models (all trained at $L$=1024) on three held-out datasets, WikiText-2 (Merity et al., 2017), the ArXiv subset of Scientific Papers (Cohan et al., 2018), and Lambada (Paperno et al., 2016): at three context lengths: a shorter-than-training length ($L$=512), the training length ($L$=1024), and a length-generalization regime ($L$=2048, double the training length). For each (model, length) configuration we report token-level zero-shot perplexity on each dataset and their unweighted average (Avg. PPL); generation perplexity (Gen PPL) computed by scoring 1024 unconditionally-sampled completions per model under a GPT-2 Large judge (Radford et al., 2019); and MAUVE (Pillutla et al., 2021) against a held-out human reference set of equal size. The $L$=1024 rows reproduce the WikiText-2, ArXiv, and Lambada columns of Table 1 (1.3B block) so each table can be read in isolation. Best result in each column is highlighted in  blue .

*Table 15.* Generation quality at $L$=512 (shorter than training length).

| Model (1.3B) | WikiText-2 | ArXiv | Lambada | Avg. PPL ↓ | Gen PPL ↓ | MAUVE ↑ |
|---|---|---|---|---|---|---|
| DiffuTran | 41.00 | 32.06 | 41.47 | 38.18 | 48.45 | 0.696 |
| DiffuMamba | 40.35 | 27.66 | 36.90 | 34.97 | 49.86 | 0.683 |
| DiffuMamba-H | **35.59** | **26.74** | **36.51** | **32.95** | **41.31** | **0.720** |

*Table 16.* Generation quality at $L$=1024 (training length). The WikiText-2, ArXiv, and Lambada columns reproduce the corresponding entries of Table 1 (1.3B block) for self-containedness.

| Model (1.3B) | WikiText-2 | ArXiv | Lambada | Avg. PPL ↓ | Gen PPL ↓ | MAUVE ↑ |
|---|---|---|---|---|---|---|
| DiffuTran | 36.63 | 23.25 | 36.73 | 32.20 | 43.06 | 0.712 |
| DiffuMamba | 34.74 | 22.98 | 36.04 | 31.25 | 42.06 | 0.704 |
| DiffuMamba-H | **31.92** | **20.67** | **34.04** | **28.88** | **37.34** | **0.744** |

*Table 17.* Generation quality at $L$=2048 (length generalization, double the training length). DiffuTran shows substantial token-level degradation (e.g., WikiText-2 PPL nearly doubles from 36.63 at $L$=1024 to 63.84 at $L$=2048), while DiffuMamba and DiffuMamba-H are essentially flat or improve on every dataset.

| Model (1.3B) | WikiText-2 | ArXiv | Lambada | Avg. PPL ↓ | Gen PPL ↓ | MAUVE ↑ |
|---|---|---|---|---|---|---|
| DiffuTran | 63.84 | 45.11 | 60.76 | 56.57 | 25.91 | 0.734 |
| DiffuMamba | 32.01 | 20.54 | 34.98 | 29.18 | 22.54 | 0.758 |
| DiffuMamba-H | **31.25** | **20.03** | **34.32** | **28.53** | **19.43** | **0.796** |

## F. Long-range retrieval (Needle-in-a-Haystack).

A natural concern with state-space backbones is that the fixed-size recurrent state could bottleneck long-range information access. To probe this directly, we run a Needle-in-a-Haystack (NIAH) passkey-retrieval evaluation adapted to masked diffusion: a single needle sentence of the form "`The magic word for {key} is: {value}`" is planted at a controlled relative depth in a haystack of unrelated text, and the model must reconstruct the masked {value} from a query placed at the end of the sequence. We sweep sequence length $L \in \{512, 1024, 2048, 4096, 8192\}$ and needle depth in $\{10\%, 25\%, 50\%, 75\%, 90\%\}$ (depth measured from the start of the haystack), with 100 random trials per cell. Table 18 reports exact-match accuracy averaged across the five depths, and Figure 5 shows the full $L \times$ depth heatmap.

The pattern in Table 18 mirrors the Avg. PPL story: at the training length all three architectures retrieve passkeys near-perfectly, but beyond training length the Mamba-based models degrade much more gracefully. The hybrid DiffuMamba-H achieves the best average retrieval at $L$=2048 (63%) and $L$=4096 (28%), suggesting that the interleaved attention layers help distribute information access across the sequence while the bidirectional Mamba layers provide the length-generalization backbone. DiffuMamba shows the strongest deep-context retrieval (Figure 5: 100% accuracy at depth 90% for $L$=2048 and 59% at depth 90% for $L$=4096), indicating that the fixed Mamba state is not the bottleneck for needles placed near the query position. At $L$=8K (8× training length), all three models effectively fail (0–1% average accuracy), bounding the regime in which length generalization holds for any of the architectures considered here.

*Table 18.* Needle-in-a-Haystack passkey-retrieval accuracy (%) at the 1.3B scale, averaged across needle depths $\{10\%, 25\%, 50\%, 75\%, 90\%\}$ with 100 trials per (length, depth) cell. All models are trained at $L{=}1024$. Within training length all three models retrieve near-perfectly. Beyond training length, `DiffuMamba` and `DiffuMamba-H` degrade much more gracefully than `DiffuTran`; at $L{=}4K$ ($4\times$ training length), `DiffuMamba-H` retains **28**% accuracy versus `DiffuTran`'s 10%. At $L{=}16K$ all three models score 0% (omitted from the table; see Figure 5).

| Sequence length | DiffuTran | DiffuMamba | DiffuMamba-H |
|---|---|---|---|
| $L{=}512$ (0.5$\times$ train) | 98% | **99%** | **99%** |
| $L{=}1024$ (1$\times$ train) | 97% | **99%** | 97% |
| $L{=}2048$ (2$\times$ train) | 47% | 56% | **63%** |
| $L{=}4096$ (4$\times$ train) | 10% | 14% | **28%** |
| $L{=}8192$ (8$\times$ train) | 0% | **1%** | 0% |

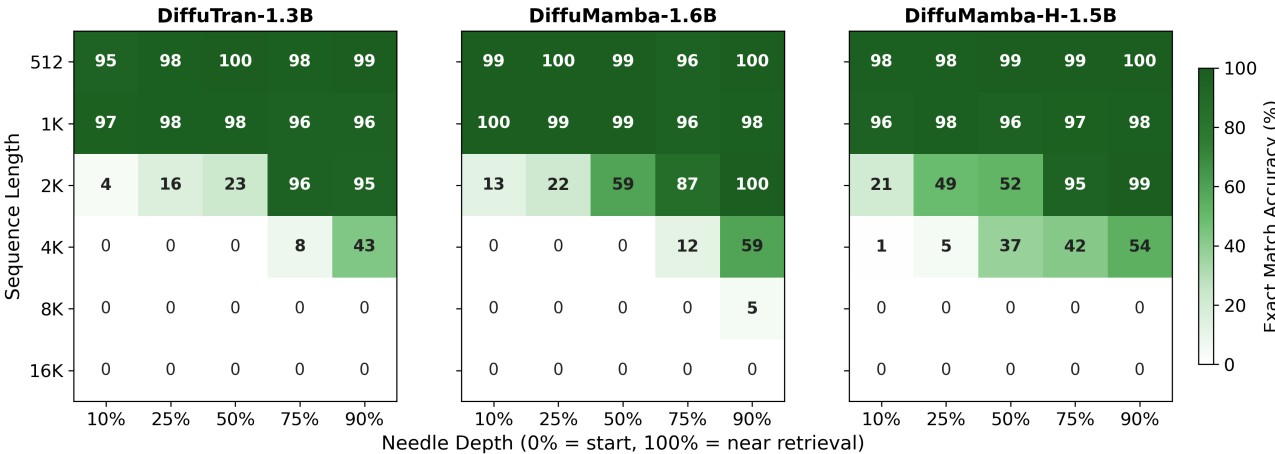

*Figure 5.* Needle-in-a-Haystack passkey-retrieval accuracy (%) across sequence lengths $\{512, 1K, 2K, 4K, 8K, 16K\}$ (rows) and needle depths $\{10\%, 25\%, 50\%, 75\%, 90\%\}$ (columns) for the three 1.3B-scale models. Each cell is exact-match accuracy averaged over 100 random trials. All models are trained at $L{=}1024$. Within training length all three architectures retrieve near-perfectly (top two rows). Beyond training length, the Mamba-based models degrade more gracefully and retain stronger deep-context (high-depth) retrieval; `DiffuMamba-H` has the most balanced cross-depth coverage at $L{=}4K$. At $L \geq 8K$ all models score $\leq 5\%$ at every depth.

**Per-depth heatmap.** Figure 5 reports the full Needle-in-a-Haystack accuracy heatmap that underlies the depth-averaged summary in Table 18. For each (sequence length $L$, needle depth) cell, accuracy is the fraction of 100 random trials in which the model exactly reconstructs the masked passkey {value}. Two patterns are visible. (i) At deeper needle positions (closer to the query at the end of the sequence), retrieval is preserved much further beyond training length: `DiffuMamba` retains 100% accuracy at depth 90% for $L{=}2048$ and 59% at depth 90% for $L{=}4096$, while `DiffuTran` drops to 95% and 43% at the same cells. (ii) At shallower depths (further from the query) the Mamba advantage persists but is smaller in magnitude. The hybrid `DiffuMamba-H` achieves the most balanced cross-depth coverage at $L{=}4096$ (1%, 5%, 37%, 42%, 54% from depth 10% to 90%), supporting the interpretation that the interleaved attention layers help distribute information access across the sequence while the bidirectional Mamba layers carry the length-generalization burden. At $L{=}8K$ and $L{=}16K$, all three models effectively fail at every depth, bounding the regime in which any of these architectures generalize on this passkey task without explicit long-context training.

## G. Limitations

While our results demonstrate strong inference gains, the joint training of diffusion language models with Mamba backbones and block cache reuse remains unexplored. Our evaluation therefore isolates the effect of backbone choice and inference algorithm on throughput and latency, but does not reflect potential gains from training models explicitly for the fastest decoding regimes. In particular, training models explicitly optimized for the fastest inference algorithms such as block diffusion with cache reuse alongside the exploration of alternative linear-time token mixers, remains a natural and immediate next step.

Further our throughput measurements assume fixed acceptance rates through the parameter $p$, which controls the number of denoising steps $K = L/p$. Although diffusion models achieve their best quality when $p = 1$, this setting effectively reduces diffusion decoding to autoregressive-style computation and defeats the primary motivation of diffusion-based generation. Our choice of moderate $p$ values reflects the intended efficiency regime of DLMs, but alternative acceptance behaviors may lead to different trade-offs.

Finally, all experiments are conducted at batch size $B = 1$. This setting is consistent with prior work (Nie et al., 2026; Wu et al., 2026) on long-context decoding and reflects the practical constraints of GPU memory at extreme sequence lengths. Evaluating larger batch sizes in the long-context regime remains an important direction for future study (a partial step in this direction is reported in Appendix D).

