# OpenReview forum: "DiffuMamba: High-Throughput Diffusion LMs with Mamba Backbone"
_ICML.cc/2026/Conference — ICML 2026 regular_

### Official Review · Reviewer_9gvB · 2026-02-28

**Soundness:** 3
**Presentation:** 3
**Significance:** 3
**Originality:** 2
**Overall Recommendation:** 3
**Confidence:** 4

**Summary:**

This paper studies the efficiency bottleneck of diffusion language models for long-context inference. It argues that using Transformers as denoisers suffers from quadratic attention cost and heavy KV-cache memory traffic, so throughput drops quickly as sequence length grows. The authors propose DiffuMamba, which replaces the Transformer denoiser backbone in a masked discrete diffusion setup with a bidirectional Mamba backbone to get linear-time sequence mixing and lower memory overhead. They also propose DiffuMamba-H, a hybrid design that inserts one attention layer every N Mamba layers to compensate for weaker global dependency modeling in purely linear mixers. They pretrain and evaluate at 240M, 0.5B, and 1.3B scale, reporting perplexity and accuracy on several commonsense reasoning benchmarks, and they provide an asymptotic complexity and latency breakdown to explain what drives throughput vs. length under different architectures and inference algorithms.

**Compliance With Llm Reviewing Policy:**

Affirmed.

**Key Questions For Authors:**

1. Why is N fixed to 5? Please provide an ablation over N with both quality and throughput curves, and discuss the minimum attention ratio needed for diffusion denoising.

2. In block-cache inference, the right-to-left state is restricted within each block. Does this significantly hurt quality, especially on long-dependency settings? Please report long-context quality as a function of block size, and quantify the gap to full-sequence denoising.

3. You claim a “systematic analysis of modern DLM inference strategies,” but the coverage still seems limited. Please either include the key missing methods or clearly justify why they are excluded, and reproduce the comparisons under the same hardware and implementation conditions.

4. At 240M scale, the pure Mamba variant is clearly worse. Does this indicate a scale threshold where diffusion denoising requires explicit global interaction? Please provide diagnostic analysis, e.g., how perplexity varies with attention-layer ratio, or error statistics across noise steps t.

**Limitations:**

Yes. The paper covers the main technical limitations.

**Strengths And Weaknesses:**

Strengths:

1. The paper pinpoints the practical bottleneck as long-context throughput and memory/IO, and supports this with a fairly systematic measurement and explanation framework.
2. It presents two clear options: a pure Mamba denoiser and a hybrid denoiser. The hybrid rule fixed-interval attention insertion is simple and easy for others to reproduce or swap into related systems.
3. The evaluation is not only about PPL and zero-shot downstream results; it also includes throughput scaling curves and multiple inference strategies, including full-sequence denoising, Fast-dLLM-style block diffusion, and block cache reuse.

Weaknesses:

1. The originality is limited. The main change is essentially “replace attention with Mamba” plus a fixed-interval hybrid, and this Transformer-to-SSM swap is already a crowded direction in both diffusion models and LMs.
2. Although the paper claims only the mixer changes, the Mamba variants often have larger parameter counts, while the MLP expansion ratio is reduced to match a parameter budget. This alters both parameterization and FLOPs structure, making it hard to attribute quality gains or throughput gains specifically to the backbone choice.
3. At 240M, DiffuMamba clearly lags the attention baseline; at 0.5B, pure Mamba is still weaker, and most of the gains come from the hybrid. This suggests pure Mamba denoisers are not reliably better, and the claims should be more cautious, e.g., hybrid works better at larger scale while pure Mamba may have a scale threshold.
4. The hybrid interval N is fixed to 5 with no systematic ablation, and there is little justification for why this frequency is optimal for diffusion denoising.

---

> ### Author Rebuttal · Authors · 2026-03-31
>
> We thank the reviewer for the detailed and constructive feedback. We address each concern below.
>
> ## W1: Addressing Limited Originality
>
> > "The main change is essentially 'replace attention with Mamba' plus a fixed-interval hybrid, and this Transformer-to-SSM swap is already a crowded direction."
>
> Regarding the lack of originality:
> - we would like to highlight that  core contribution is not the swap itself,  it is **identifying the recipe that unlocks DLM efficiency**: DiffuMamba-H + block-cache is the only configuration that scales linearly with L while matching Transformer quality (Figures 2c-d). This provides a clear recipe for future efficient long-context DLMs.
>
> - The throughput analysis framework (Table 3 + Figure 3) provides the first unified comparison of FLOPs, memory, arithmetic intensity, and throughput across 5 DLM inference strategies,  a tool the community can reuse independently of our specific architectures:
>
> ---
> ## W2: Addressing Parameter Count Mismatch with Flop Analysis
>
> > "Although the paper claims only the mixer changes, the Mamba variants often have larger parameter counts,"
>
> Please kindly refer to the rebuttal reply given to **qvWt** for W1
>
>
> ---
> ## W3: Addressing Cautious Claims on 240M Underperformance
> > "At 240M, DiffuMamba clearly lags the attention baseline... claims should be more cautious."
>
> We agree the claims should be nuanced. However, we emphasize that similar behavior has also been observed in the SSM literature:
>
> - Waleffe et al. (2024) show that at early training (1.1T tokens), Mamba and Mamba-2 at 8B produce nearly 15 points lower MMLU accuracy than a Transformer, and while more compute narrows the gap, it does not fully close it for all tasks, suggesting SSMs require more capacity to achieve parity.
> - Mamba-3 (ICLR 2026) highlights SSMs need sufficient model capacity (state size relative to hidden dimension) to effectively compress sequence information.
>
> The scaling trend is the key signal: from 240M to 1.3B, DiffuMamba goes from marginally behind to clearly ahead of DiffuTran on validation PPL (Table 2: 21.41 vs. 22.72 at Quokka budget). This improving trajectory is more informative than any single scale point.
>
> ---
> ## W4: Providing Ablations for interleaved attention layers (N) in Hybrid models.
>
> > "The hybrid interval N is fixed to 5 with no systematic ablation."
>
> Please kindly refer to the rebuttal reply given to wqN4 in W3.
>
> ---
> ## Q1: Providing Ablations for interleaved attention layers (N) in Hybrid models.
>
> Addressed in W4 above.
>
> ---
> ## Q2: Providing Block-Cache Quality for Long sequence lengths
> > "In block-cache inference, the right-to-left state is restricted within each block. Does this significantly hurt quality?"
>
> We  evaluated Gen PPL at G={32, 64, 128} on long sequences as shown here:
>
> https://anonymous.4open.science/api/repo/Subnet-39B5/file/figures/block_cache_results.pdf?v=234e4267
>
> We observe that block-cache does not significantly hurt quality when the block size is matched to the architecture.
>
> - DiffuMamba achieves the best overall Gen PPL (12.09 at L=16K, G=128). Larger blocks help because the forward state accumulates more context before the within-block reverse scan operates.
> - DiffuMamba-H is best at G=64 (14.63 at L=8K, 14.58 at L=16K), where the interleaved attention complements Mamba's recurrence at moderate block sizes.
> - DiffuTran is the most stable across block sizes (19.60–22.61 range) but never achieves the best Gen PPL at any configuration. We hypothesize that its KV-cache recomputation per block insulates it from block size effects, at the cost of higher per-block compute.
>
> ---
> ## Q3: Clarifying Systematic Analysis Coverage
> > "Coverage still seems limited. Please either include the key missing methods or clearly justify why they are excluded."
>
> Our analysis covers 5 distinct inference configurations across 3 backbone architectures (15 combinations total).
>
> We would like to request the reviewer to please kindly clarify which methods in their opinion are missed.
>
> ---
> ## Q4: Addressing Scale Threshold
>
> > "Does this indicate a scale threshold where diffusion denoising requires explicit global interaction?"
>
> According to our understanding, the 240M DiffuMamba performs slightly worse than DiffuTran because of the capacity constraint in Mamba in the masked diffusion setting. Mamba compresses sequence context into a fixed-size recurrent state.
>
> We hypothesize that at this scale, the state capacity (d_state x d_inner) is insufficient to faithfully represent the full bidirectional context needed for masked token reconstruction in diffusion modelling, whereas attention provides exact pairwise interactions regardless of model size. As scale increases to 1B+, the larger hidden dimensions and deeper layer stack allow the recurrent state to build richer bidirectional representations, closing the gap with attention-based models.

---

> > ### Author Rebuttal · Reviewer_9gvB · 2026-04-02
> >
> > Thank you for the rebuttal and the additional block-cache quality results across block sizes.

---

> > > ### Author Response · Authors · 2026-04-02
> > >
> > > We are glad that we could **resolve** all your concerns **adequately**. We would like to humbly ask you if you would be willing to upgrade your score.
> > >
> > > Thanks.

---

### Official Review · Reviewer_wqN4 · 2026-03-10

**Soundness:** 3
**Presentation:** 3
**Significance:** 3
**Originality:** 3
**Overall Recommendation:** 4
**Confidence:** 3

**Summary:**

This paper addresses the key limitation of diffusion language models under long sequence lengths, where the quadratic complexity of Transformers becomes a bottleneck. The authors propose the DiffuMamba architecture, which introduces bidirectional Mamba-2 into the discrete masked diffusion process, enabling linear-time scaling of inference with respect to sequence length. In addition, the proposed hybrid variant DiffuMamba-H not only matches the performance of Transformers at the 1.3B scale, but also achieves significant throughput improvements.

**Compliance With Llm Reviewing Policy:**

Affirmed.

**Key Questions For Authors:**

See the weaknesses above.

**Limitations:**

yes

**Strengths And Weaknesses:**

Strengths:
This work is the first to introduce state space models (SSMs) into the discrete masked diffusion process, achieving linear-time inference scaling. The idea demonstrates clear novelty.

Weaknesses:
The evaluation in this work is limited to relatively small-scale models (up to 1.3B parameters), which may not fully reflect the scaling dynamics inherent in large-scale diffusion language modeling. To convincingly demonstrate that DiffuMamba is a reliable alternative to Transformer-based diffusion models (e.g., LLaDA), the authors need to show that the efficiency–quality trade-off observed at the 1.3B scale still holds at larger scales where global dependency modeling becomes more critical.

Introducing Mamba may also make training more challenging, since the parallelization of bidirectional recurrent structures is typically more complex than highly optimized Transformer operators (such as FlashAttention-3), and may introduce numerical stability issues.

The paper adopts a hybrid design where one attention layer is inserted every five Mamba layers, but no ablation study is provided

---

> ### Author Rebuttal · Authors · 2026-03-31
>
> We thank the reviewer for recognizing DiffuMamba as the first SSM-based discrete masked diffusion model and for the constructive feedback. We believe the additional results and clarifications below address the reviewer's concerns and would appreciate it if the reviewer would consider a corresponding increase in the score.
>
> ## W1: Limited to 1.3B Scale
> > "The evaluation is limited to relatively small-scale models (up to 1.3B parameters), which may not fully reflect the scaling dynamics inherent in large-scale diffusion language modeling."
>
> The efficiency advantage is structural and scale independent. Our asymptotic analysis (Table 3) shows DiffuTran's per-token cost is O(d² + Ld) while DiffuMamba's is O(d² + d·d_state), independent of L. At L=65K this means DiffuTran spends 83% of its FLOPs on the attention term alone, while DiffuMamba's cost stays flat. This gap can only grow at longer sequences.
>
> On the quality side, the trend from 240M to 1.3B is monotonically favorable: DiffuMamba-H goes from matching DiffuTran to 11% better Val PPL (20.17 vs 22.72). SSMs have been validated at much larger scales in the AR domain (Jamba at 52B, Falcon-Mamba at 7B), and as DLMs scale further (LLaDA2.0 at 100B), the quadratic attention bottleneck that DiffuMamba eliminates becomes the dominant cost.
>
> ---
> ## W2: Training Challenges with Bidirectional Mamba
>
> > "Parallelization of bidirectional recurrent structures is typically more complex than FlashAttention-3, and may introduce numerical stability issues."
>
> We appreciate this concern, but we think it is based on a misconception about how Mamba-2 training works. We want to clarify:
>
> **Mamba-2 is NOT a sequential RNN during training.** Mamba-2 uses the **Structured State Space Duality (SSD)** framework (Dao & Gu, 2024), which reformulates the state space computation as a semi-separable matrix multiplication. This enables hardware-efficient parallel chunked scans,  the same level of parallelism as attention during training.
>
> **Numerical stability:** We observed zero numerical stability issues across all training runs (3 scales x 3 architectures = 9 full training runs). No NaN gradients, no loss spikes, no divergence events. The training loss curves for DiffuMamba are as smooth as those for DiffuTran.
>
> ---
> ## W3: Providing Ablation Study for N=5
>
> > "One attention layer is inserted every five Mamba layers, but no ablation study is provided."
>
> We chose N=5 from Nemotron-H (Blakeman et. al. 2025), which had 1 attention layer per 5 mamba layers in each block. For the 1.3B model, with 24 layers, we chose N=5 to have similar alignment. Below we trained DiffuMamba-H at 0.5B scale with every N $\in$ {3, 5, 9, 15} plus pure Mamba and pure Attention models, all under identical training budgets (50B tokens, same hyperparameters).
>
> We report Gen PPL (fluency via GPT-2 Large) and MAUVE (distributional similarity to human text) on 512 unconditional samples:
>
> | N | Attn% | Gen PPL (↓) | MAUVE (↑) |
> |---------|-------|-------------|-----------|
> | pure attention | 100% | 50.97 | 0.672 |
> | 3 | 33% | 48.87 | 0.721 |
> | 5 (default) | 20% | 49.37 | 0.719 |
> | 9 | 11% | 50.33 | 0.724 |
> | 15 | 7% | 49.56 | 0.681 |
> | pure mamba | 0% | 48.07 | 0.674 |
>
> Three observations:
>
> 1. Every hybrid outperforms both pure endpoints on MAUVE. The two mixer types are genuinely complementary: Mamba contributes fluency, and attention contributes distributional diversity.
> 2. The quality plateau is remarkably flat. Gen PPL varies only 48.07–50.97 (6% range) across the entire spectrum from pure Mamba to pure attention. N=5 is a robust default, not a fragile optimum.
> 3. N=5 sits at a good efficiency-quality tradeoff. It achieves Gen PPL 49.37 and MAUVE 0.719 (within 3% of best on both) while using only 7 out of 36 attention layers, which directly reduces inference FLOPs at long sequences.

---

> > ### Author Rebuttal · Reviewer_wqN4 · 2026-04-02
> >
> > Thank the authors for their detailed responses to my concerns in the rebuttal. The authors have provided fairly thorough clarifications addressing each of the three main issues I raised.

---

> > > ### Author Response · Authors · 2026-04-02
> > >
> > > We are glad that we could provide **thorough clarification** addressing **each** of the issues you requested, thus **fully resolving** your concerns. We would like to humbly ask you if you would be willing to upgrade your score.
> > >
> > > Thanks.

---

### Official Review · Reviewer_wSbE · 2026-03-12

**Soundness:** 3
**Presentation:** 3
**Significance:** 3
**Originality:** 2
**Overall Recommendation:** 4
**Confidence:** 3

**Summary:**

This paper introduces DiffuMamba, a dLLM adopting the Mamba architecture with linear-time sequence modeling characteristics. DiffuMamba matches the accuracy of standard transformer based dLLMs and scales up to 1.3B parameters. DiffuMamba achieves at most 8.2x speedup of inference throughput, especially on long sequences.

**Compliance With Llm Reviewing Policy:**

Affirmed.

**Key Questions For Authors:**

1. Why does the pure DiffuMamba model sometimes underperform the standard Transformer baseline at the smallest scale of 240M? Does this suggest the architecture is unstable regarding the performance?
2. How does the throughput of DiffuMamba change at greater batch sizes?

**Limitations:**

yes

**Strengths And Weaknesses:**

**Strengths**

1. By interleaving attention layers with Mamba layers, the hybrid model successfully balances efficient generation with the capturing of global semantic dependencies.
2. The empirical evaluation of Mamba-layer design is sound and comprehensive. The authors evaluate the proposed architecture at multiple scales and demonstrate its effectiveness.

**Weakness**

1. The proposed model only scales to 1.3B, which is smaller than the common dLLMs of 8B and greater sizes (e.g., 30B). It is unclear if the efficiency and quality benefits of DiffuMamba still holds at greater scales.
2. The zero-shot downstream task scores for the 1.3B models are quite low for all models. It is unclear whether the reasoning capability of Transformer architecture is effectively exploited.
3. The evaluations on inference speed are all conducted at a batch size of 1\. More evaluations on greater batch sizes would make the result more comprehensive and persuasive.

---

> ### Author Rebuttal · Authors · 2026-03-31
>
> We thank the reviewer for the constructive feedback. We address each point below.
>
> ## W1: Efficiency and Quality Beyond 1.3B
> > "The proposed model only scales to 1.3B, which is smaller than the common dLLMs of 8B and greater sizes. It is unclear if the efficiency and quality benefits still hold at greater scales."
>
> The efficiency advantage follows directly from asymptotic analysis in Table 3: DiffuTran's per-token cost grows as O(d² + Ld) while DiffuMamba's is O(d² + d·d$_{state}$), independent of L. As sequence length grows, the 4Ld attention term dominates (83% of DiffuTran's FLOPs at L=65K), while DiffuMamba's cost stays flat. This gap only grows at longer contexts. At sizes beyond 1.3B caching/ cache recomputation are more expensive making Mamba-DLMs more attractive.
>
> As for quality, the trajectory from 240M to 1.3B is monotonically improving (DiffuMamba-H goes from matching DiffuTran to 11% better Val PPL). This is consistent with Jamba (52B), Falcon-Mamba (7B) -- all show SSM quality improves with scale. As DLMs scale (LLaDA2.0 at 100B), quadratic attention bottleneck becomes the dominant cost increasing linear-time backbones' advantage.
>
> ---
> ## W2: Low Zero-Shot Downstream Task Scores
> > "The zero-shot downstream task scores for the 1.3B models are quite low for all models."
>
> The absolute scores are low because these are base models without instruction tuning or few-shot prompting. This is expected and consistent with the literature:
>
> - LLaDA-8B base (Nie et al., 2025) also reports modest raw downstream scores before supervised fine-tuning (SFT). It is only after SFT that LLaDA-8B matches LLaMA-3 performance.
>
> - GPT-2 1.5B, TinyLlama (Zhang et al., 2024) achieves similar zero-shot accuracy ranges on these benchmarks (25-35% on ARC, OBQA).
>
> These benchmarks (HellaSwag, ARC, etc) are designed to be challenging even for much larger models.
>
> **The key metric is the relative comparison:** DiffuMamba-H outperforms DiffuTran by **~4%** average accuracy (38.24 vs. 33.81, Table 4). This consistent improvement across all five tasks demonstrates that the Mamba backbone provides a genuinely stronger denoising signal that transfers to downstream reasoning. The improvement is particularly notable on OpenBookQA (+7.3%) and LogiQA (+10.5%), both of which require multi-hop reasoning.
>
> ----
> ## W3: Addressing Throughput Evaluations at Batch Size > 1
> > "More evaluations on greater batch sizes would make the result more comprehensive and persuasive."
>
> We provide throughput plots at batch sizes B = 8 for all three 1.3B models under both full-sequence denoising and fast-dLLM caching, at sequence lengths L in {64, 256, 1024, 2048, 8192, 16384}  We observed OOM error beyond 16K. But the overall trend is very similar to the figures 2a and 2b. Here are the results
>
> https://anonymous.4open.science/api/repo/Subnet-39B5/file/figures/throughput_rebuttal_B8.pdf?v=c7174678
>
> **Observations:**
> The overall trend is similar to Fig 2a and Fig 2b, where the throughput is comparable for all the methods for short sequences.
>
> - DiffuMamba gives **3.5x** improvement over Diffutran, and the hybrid variant gives **2.0x** improvement at the sequence length of 16K.
> - DiffuMamba-H coupled with Fast-dLLM has the maximum throughput at 16K sequence length around 256, with an improvement of **2.2x** over DiffuMamba-H without caching.
>
> ---
> ## Q1: Why Does Pure DiffuMamba Underperform at 240M?
> > "Does this suggest the architecture is unstable regarding the performance?"
>
> No, this does not indicate architectural instability. Rather, it reflects a capacity constraint of Mamba in the masked diffusion setting. Mamba compresses sequence context into a fixed-size recurrent state. At 240M scale, this compression must simultaneously support both forward and backward context integration (via our antiparallel scan design) and the denoising objective, which is  a more demanding task than causal language modeling.
>
> - We hypothesize that at this scale, the state capacity is insufficient to faithfully represent the full bidirectional context needed for masked token reconstruction in diffusion modelling, whereas attention provides exact pairwise interactions regardless of model size.
>
> - As scale increases to 1B+, the larger hidden dimensions and deeper layer stack allow the recurrent state to build richer bidirectional representations, closing the gap with attention-based models.
>
> - Crucially, DiffuMamba-H at 240M already matches DiffuTran (Table 2: 42.67 vs. 43.82 Chinchilla, 32.49 vs. 32.11 Quokka). This shows that even a small amount of attention (~20%) fully compensates for the SSM state bottleneck at small scales, confirming this is a capacity issue rather than architectural instability.
>
> **Training stability:** We observed no numerical issues (NaN gradients, loss spikes, divergence) in any DiffuMamba training run across all three scales. The training loss curves are smooth and well-behaved.
>
> ---
> ## Q2: Throughput at Greater Batch Size
> Addressed in W3 above.

---

> > ### Author Rebuttal · Reviewer_wSbE · 2026-04-04
> >
> > None

---

### Official Review · Reviewer_zdMp · 2026-03-12

**Soundness:** 2
**Presentation:** 2
**Significance:** 3
**Originality:** 2
**Overall Recommendation:** 4
**Confidence:** 3

**Summary:**

This paper introduces DiffuMamba, a masked discrete diffusion language model that replaces standard Transformer blocks with Mamba2 layers. The primary goal of the paper is to resolve the efficiency challenge in diffusion LMs due to the quadratic computation cost caused by self-attention. The models are trained and evaluated at three scales (240B, 0.5B, and 1.3B) on the DCLM dataset, achieving competitive perplexity with substantially higher inference throughput compared with transformer baselines.

**Compliance With Llm Reviewing Policy:**

Affirmed.

**Final Justification:**

Thank you for the new experimental analysis. My concerns have largely been addressed, and I will raise my score accordingly. However, after reviewing the other reviewers’ comments, I still suggest that the authors further highlight the originality of the paper (as Reviewer 9gvB also noted). The authors’ response to this point seems a bit insufficient. Additionally, I recommend training on longer sequences to better demonstrate the performance gap between DiffuMamba and the Transformer.

**Key Questions For Authors:**

1. The evaluation is conducted on general-domain text benchmarks, while structured generation tasks such as code and mathematical reasoning place higher demands on long-range modeling capacity. Therefore, could authors give additional experiments on benchmarks like HumanEval and GSM8K? I believe the results would give more insights to the community.
2. Are there any insights or challenges when applying the Mamba architecture to diffusion LMs? I believe this could help enhance the contribution of the paper.

**Limitations:**

yes

**Strengths And Weaknesses:**

* Strengths
1. At the size of 1.3B, the proposed DiffuMamba-H can achieve lower validation perplexity than Diffusion Transformers under different compute budgets. Furthermore, experiments show that DiffuMamba can match or even improve upon MHA-based DLMs in validation and zero-shot PPL with higher efficiency across different model scales.
2. DiffuMamba shows throughput advantages on long sequences compared with transformer-based models. And the paper also provides a detailed theoretical analysis of FLOPs, memory, and computational intensity to support the experimental results.
* Weaknesses
1. As shown in Table 5, DiffuMamba has more parameters than Diffusion Transformers (e.g. DiffuMamba-1.6B has 23% more parameters than DiffuTrans-1.3B). However, it seems that tabs. 1 and 2 present perplexity with the idea that they are equal-scale, resulting in a discrepancy between Tab. 5 and them. Although the model size difference is small, it is recommended to present consistent settings in the paper. Additionally, it would be better to provide an experiment to verify that the gain of DiffuMamba is attributed to intrinsic architecture improvement instead of more parameters.
2. Figure 2 shows that the DiffMamba can achieve higher throughput than Diffusion Transformers across a wide range of sequence lengths. However, the generation quality (e.g., validation perplexity, MAUVE) at different seq lengths is not provided. It would be helpful if the authors could also report these quality metrics to enable a more comprehensive comparison of the efficiency–performance trade-off between DiffMamba and Diffusion Transformer-based models.
3. Token-level perplexity alone may be insufficient to characterize generation quality. Perplexity measures token distribution fit but does not capture fluency, coherence, or diversity of actual samples. It would be better to additionally report MAUVE or generative perplexity evaluated by an external model.

---

> ### Author Rebuttal · Authors · 2026-03-31
>
> ## W1: Addressing Parameter Count Discrepancy via Flop Analysis
>
> > "DiffuMamba has more parameters than DiffuTran (e.g. 1.6B vs 1.3B). It would be better to verify that the gain is attributed to intrinsic architecture improvement instead of more parameters."
>
> We provide a detailed FLOPs-per-token analysis computed directly from the checkpoint weight shapes, validated theoretically:
>
> https://anonymous.4open.science/api/repo/Subnet-39B5/file/figures/flops_table.pdf?v=cceceb96
>
> Key findings:
>
> - **More params ≠ more FLOPs.** Mamba's extra params are in linear projections (in_proj: [8056, 1920]) that cost O(d²) per token,  same as attention's QKV ([5760, 1920]). Attention then adds **4Ld FLOPs with zero parameters** (Q@K^T + Attn@V), which dominates at long sequences.
>
> ---
> ## W2: Providing Generation Quality (Gen PPL & Mauve scores) at Different Sequence Lengths
>
> > "The generation quality (e.g., validation perplexity, MAUVE) at different seq lengths is not provided."
>
> We provide a comprehensive evaluation of generation quality across sequence lengths, including Gen PPL (fluency via GPT-2 Large) and MAUVE (distributional similarity to human text):
>
>
> https://anonymous.4open.science/api/repo/Subnet-39B5/file/figures/mauve_table.pdf
>
> Key findings:
>
> - **DiffuMamba-H achieves the best Val PPL, Gen PPL, and MAUVE** at all three sequence lengths.
> - At L=2048, DiffuTran's zero-shot PPL degrades by +76% (32.20 → 56.57 avg).
> - **DiffuMamba-H is stable at L=2048.** Val PPL changes only -1.2% (28.88 → 28.53). Mamba's recurrence processes each token identically regardless of position, no extrapolation needed.
>
> ---
> ## W3: Addressing Perplexity Alone Is Insufficient
>
> > "Token-level perplexity alone may be insufficient. Report MAUVE or generative perplexity evaluated by an external model."
>
> Addressed in W2
>
>
> ---
> ## Q1: Addressing HumanEval and GSM8K Benchmarks
>
> > "Could authors give additional experiments on benchmarks like HumanEval and GSM8K?"
>
> We would like to clarify that these are extremely challenging benchmarks for 1.3B base models without instruction tuning. For context, these benchmarks are known to require either
>
> - large-scale instruction tuning on code/math data, or
> - model sizes well above our 1.3B parameter budget, to achieve meaningful scores.
>
> Base-model autoregressive LMs at smaller scales, e.g., LLaMA 2 7B Base, achieving only 13.1 on GSM8K and 12.8 on HumanEval [Nie et. al. LLaDA, 2025] perform poorly without instruction tuning. As a diffusion LM trained without math or code-specific SFT data, we do not expect competitive performance on these benchmarks.
>
> ---
> ## Q2: Insights and Challenges for Mamba in Diffusion LMs
>
> > "Are there any insights or challenges when applying the Mamba architecture to diffusion LMs?"
>
> The main takeaways from our experience:
>
> 1. **Bidirectionality is essential, with simple additive fusion** We tried concatenation and gated fusion; additive merge (Eq. 6) performed better while being simplest. The two scan directions learn complementary features that combine without interference.
>
> 2. **Mamba is a natural fit for block-cache inference.** Its fixed-size state (O(d_state × d_inner)) is a compressed summary of all past context. After denoising one block, the forward state carries over to the next block with no recomputation. Attention requires either storing the full KV-cache (O(L × d)) or recomputing it per block (Fast-dLLM). This is why DiffuMamba + Block Cache achieves the best throughput in Figures 2c/2d.
>
> 3. **No positional encoding is needed.** Mamba encodes ordering through its recurrence. This eliminates RoPE and, as our length generalization results show, lets the model handle L=2048 without the degradation that attention suffers when extrapolating beyond trained positions.
>
> **Challenges:**
>
> 1. **No SSM state reuse compatibility for recent KV caching methods like Fast-dLLM.** In full-sequence denoising, the input changes every step, so Mamba must recompute from scratch. Attention can partially reuse KV-cache for unchanged tokens. Block-cache inference (Section 4.3) sidesteps this by caching at block boundaries.
>
> 2. **The hybrid ratio (N) is a new hyperparameter.** Our N-ablation (see response to 9gvB W4) shows a broad plateau, but future work could explore noise-level-adaptive mixing, e.g., more attention at high noise (sparse context) and more Mamba at low noise (dense context).

---

> > ### Author Rebuttal · Reviewer_zdMp · 2026-04-04
> >
> > I thank the authors for the rebuttal; most of my concerns are addressed. I fully understand the concerns regarding Q1. Actually, my intention was to see the model's long-range capability. Given Mamba’s known limitation due to its fixed state size, a simple 'Needle In A Haystack' or 'Passkey' test would be very helpful to demonstrate DiffuMamba's effectiveness in long-context tasks compared to Transformers.

---

> > > ### Author Response · Authors · 2026-04-06
> > >
> > > We are glad that we were able to resolve most of the reviewer's concerns. We thank you for the suggestion on long-range capability, and are happy to provide Needle-in-a-Haystack (NIAH) results comparing DiffuMamba and DiffuMamba-H against DiffuTran.
> > >
> > >
> > > ## Discussion
> > >
> > > We would like to first clarify a nuance in the framing of the fixed-state-size concern. We emphasize that Mamba's state is not a passive buffer, it is **selectively** updated at every position via input-dependent gating. Dao & Gu (2024) further show through **Structured State Space Duality (SSD)**, that Mamba-2's recurrence is mathematically equivalent to a form of structured attention, establishing that SSMs have equivalent representational capacity for sequence modelling. Empirically, Jamba (Lieber et al., 2024) demonstrates strong performance at 256K context length using hybrid Mamba-attention at 52B scale, confirming that fixed-size states do not bottleneck long-range modeling in practice.
> > >
> > > ## Results on Needle in a Haystack (NIAH):
> > >
> > > ### Methodology
> > >
> > > We adapt NIAH for masked diffusion: a needle (*"The magic word for {key} is: {value}"*) is planted at a controlled depth in a long sequence, and the model must reconstruct the masked {value} in a query placed at the end. We sweep sequence length L ∈ {512, 1024, 2048, 4096, 8192}, depths ∈ {10%–90%}, 100 trials each. All models were trained on L=1024.
> > >
> > > Here is the full exact-match accuracy heatmap comparison at various sequence lengths, in the link below:
> > >
> > > https://anonymous.4open.science/api/repo/Subnet-39B5/file/figures/niah_heatmap_all.png?v=a1c1f302
> > >
> > > | Seq Length | DiffuTran | DiffuMamba | DiffuMamba-H |
> > > |-----------|-----------|------------|--------------|
> > > | 512 (0.5×) | 98% | **99%** | 99% |
> > > | 1024 (1× train) | 97% | **99%** | 97% |
> > > | 2048 (2×) | 47% | 56% | **63%** |
> > > | 4096 (4×) | 10% | 14% | **28%** |
> > > | 8192 (8×) | 0% | 1% | 0% |
> > >
> > > *Values are exact-match accuracy averaged across all 5 needle depths (10%–90%), 100 trials each.*
> > >
> > > These results highlight the capability of Mamba and Hybrids for diffusion modelling for long range capabilities.
> > > - At training length (≤1024), all models retrieve near-perfectly.
> > > - Beyond training length, **Mamba-based models degrade significantly more gracefully than the pure Transformer baseline.**
> > > - At L=4096 (4× training length), **DiffuMamba-H retains 28% average accuracy** where DiffuTran drops to **10%**.
> > >
> > > DiffuMamba itself shows the **strongest near-retrieval performance (59% at depth 90%)** at L=2048, while DiffuMamba-H achieves the **broadest coverage across needle depths.**
> > >
> > > DiffuMamba-H also shows the most balanced retrieval at L=4096. This suggests the interleaved attention layers help distribute information access, while the bidirectional Mamba layers provide the length-generalisation backbone, further evidence that the fixed state is not the bottleneck, but a structural asset.
> > >
> > > We hope these results and the accompanying clarification address the reviewer's concern for long range capability.

---

### Decision · Program_Chairs · 2026-04-30

**Decision:**

Accept (regular)

**Comment:**

The authors present a masked discrete diffusion language model that replaces the standard quadratic-cost Transformer backbone with bidirectional Mamba-2 layers to achieve linear-time sequence modeling. To balance efficiency with global dependency modeling, the authors also propose a hybrid variant which interleaves attention layers at fixed intervals within the Mamba stack. Evaluated at scales up to 1.3B parameters, the architecture demonstrates competitive perplexity and inference throughput compared to Transformer baselines, particularly on long sequences. The work further provides a theoretical and empirical analysis of various inference strategies.

The reviewers found the topic important and relevant to today's research efforts on DLLMs, and were convinced by the method's efficiency and performance. There were some issues raised in the initial reviews:
- Evaluating the model only up to 1.3B parameters, whereas modern DLLM architectures are often scaled to ~30B parameters
- The proposed method uses more parameters than the baseline Diffusion Transformers which could make an unfair comparison
- The proposed method underperformed the standard Transformer baseline at the 240M scale
- Lack of an ablation study to justify the fixed hybrid interval design
- Token-level perplexity is insufficient as a generation quality metric
- Inference speed tested at BS=1 only.
- The reviewers disagreed about the level or novelty/originality involved in incorporating Mamba into DLLMs

Generally these issues were resolved. The reviewer who left their score as "Weak Reject" also noted "all concerns resolved".
- While the authors did not provide new results for models beyond 1.3B, they discussed the performance trends they observed
- Param count mismatches was acknowledged, and a new study produced to examine FLOPs per token
- Underperformance at 240M was acknowledged and attributed to the more difficult information compression task that Mamba does, which requires larger architectures to take advantage of. This is not a fundamental reason to reject the method.
- The choice of hybrid interval setting was explained as coming from Nemotron, and an ablation was done with new results
- The MAUVE metric was added
- BS=8 was tested
- There can be some disagreement on originality, but given that the authors have appropriately acknowledged prior work, this is not a fundamental issue.

Since reviewers agreed the work is sound, I am recommending acceptance.

The authors must incorporate the new studies that address reviewer comments into their final version of the work.

---
Note to authors - One reference in your paper appears to be hallucinated or incorrectly entered:

Gu, A. and Goel, T. D. Mamba: Linear-time sequence modeling with selective state spaces. arXiv preprint
arXiv:2312.00752, 2023.

Authors who use AI tools are responsible for checking the quality of outputs. Hallucinated references are not acceptable.